# A Cas3-base editing tool for targetable in vivo mutagenesis

Anna Zimmermann [1,2], Julian E. Prieto-Vivas [1,2], Charlotte Cautereels[1,2], Anton Gorkovskiy [1,2], Jan Steensels[1,2], Yves Van de Peer [3,4,5,6] & Kevin J. Verstrepen [1,2]

The generation of genetic diversity via mutagenesis is routinely used for protein engineering and pathway optimization. Current technologies for random mutagenesis often target either the whole genome or relatively narrow windows. To bridge this gap, we developed CoMuTER (Confined Mutagenesis using a Type I-E CRISPR-Cas system), a tool that allows inducible and targetable, in vivo mutagenesis of genomic loci of up to 55 kilobases. CoMuTER employs the targetable helicase Cas3, signature enzyme of the class 1 type I-E CRISPR-Cas system, fused to a cytidine deaminase to unwind and mutate large stretches of DNA at once, including complete metabolic pathways. The tool increases the number of mutations in the target region 350-fold compared to the rest of the genome, with an average of 0.3 mutations per kilobase. We demonstrate the suitability of CoMuTER for pathway optimization by doubling the production of lycopene in *Saccharomyces cerevisiae* after a single round of mutagenesis.

Random mutagenesis is a powerful technique in biological research used for applications ranging from genotype–phenotype associations and lineage tracing to protein engineering and directed evolution. One prime field of application for random mutagenesis is the optimization of microbial cell factories. Here, complex heterologous pathways are introduced in microbial cells to produce specific compounds, such as enzymes, lipids, antibodies, and other complex biomolecules[1–3]. Introduction and expression of these complex metabolic pathways commonly lead to problems such as inefficient fluxes, low enzymatic activity, and/or build-up of (toxic) intermediates and therefore require substantial optimization to obtain desirable production yields.

Traditional techniques for random mutagenesis use exogenous mutagens like UV radiation or chemicals like EMS and MMS that act on a genome-wide level[4–6]. While these approaches result in diverse genetic landscapes, they often generate random mutations throughout the genome that reduce cellular fitness. Myriad approaches have

been developed to overcome these limitations by generating genetic diversity in a more targeted way, both in vitro[7–14] and in vivo[15–24].

An early example of in vitro generation of diversity is multiplex automated genome evolution (MAGE). MAGE iteratively introduces in vitro generated oligonucleotides from a vast pool that anneal to the lagging strand of the replication fork to produce targeted substitutions, insertions, and deletions in a semi-automated protocol[7]. While MAGE and other in vitro mutagenesis techniques have proven extremely useful in some cases, they are limited by several factors, including the need to generate oligonucleotide libraries in vitro followed by transformation of the respective host, low mutation rates, the need for complex automation, and most importantly, limited reach/target window.

Recent advances in targeted, in vivo mutagenesis often rely on Cre recombinase[25,26] or combinations of CRISPR-Cas9-variants, error-prone DNA/RNA polymerases, and DNA deaminases[16–21,23,27,28]. EvolvR[15] and CRISPR-X[27] are two examples that rely on Cas9 targeting

[1]VIB Laboratory for Systems Biology, VIB-KU Leuven Center for Microbiology, Leuven 3001, Belgium. [2]Laboratory for Genetics and Genomics, Center of Microbial and Plant Genetics, Department M2S, KU Leuven, Gaston Geenslaan 1, 3001 Leuven, Belgium. [3]Department of Plant Biotechnology and Bioinformatics, Ghent University, Ghent, Belgium. [4]VIB Center for Plant Systems Biology, Ghent, Belgium. [5]Department of Biochemistry, Genetics and Microbiology, University of Pretoria, Pretoria, South Africa. [6]College of Horticulture, Academy for Advanced Interdisciplinary Studies, Nanjing Agricultural University, 210095 Nanjing, China. ✉e-mail: yves.vandepeer@psb.ugent.be; kevin.verstrepen@kuleuven.be

and employ a DNA polymerase and cytidine deaminase, respectively. Though these systems show great utility in bacteria, yeast, and mammalian cells, they are constrained to operating in narrow windows of ~40 bp around the gRNA target site, making them inadequate to mutagenize expansive genomic regions. In an alternative strategy, Cravens et al. used TaRgeted In vivo Diversification ENabled by T7 RNAP (TRIDENT) to introduce mutations in any DNA sequence downstream of a T7 promoter. TRIDENT consists of a fusion of T7 RNA polymerase to a cytidine or adenine deaminase as well as additional DNA repair factors that further increased the mutational diversity[17]. Though TRIDENT allows the continuous mutagenesis of a ~2 kb region, its action radius is insufficient to cover multiple genes in a metabolic pathway. Despite the wealth of available approaches for random mutagenesis, there is currently no tool available that allows inducible, continuous in-vivo mutagenesis of specific, targetable regions in the native genome while at the same time having a sufficiently large activity window to encompass complex metabolic pathways without affecting the rest of the genome.

Here we describe Confined Mutagenesis using a Type I-E CRISPR-Cas system (CoMuTER), a tool for random mutagenesis within long and defined genomic regions. The tool is based on the combination of Cas3 from the class 1 type I-E CRISPR-Cas system of *E. coli*[29], and a cytidine deaminase. Class 1 CRISPR-Cas systems represent ~90% of CRISPR-Cas loci in nature[29,30] and offer unique attributes when compared to the canonical class 2 systems featuring signature enzymes like Cas9 or Cas12[31,32]. Notably, they feature inherent guide RNA processing[33], higher target site specificity, and new application possibilities linked to the activity of Cas3, a dual nuclease, and helicase enzyme[34–36]. All class 1 systems use a multi-subunit effector complex called CRISPR-associated complex for antiviral defense (Cascade) that is responsible for pre-CRISPR RNA processing, crRNA binding, and target site recognition[37,38]. The activity of Cascade and Cas3 together identifies, unwinds (>10 kb)[35,36], and intermittently cuts the resultant ssDNA from the invading species. We hypothesized that fusing Cas3 to a cytidine deaminase, which requires single-stranded DNA as a substrate, would allow for the introduction of random deaminations over a large, multi-kilobase region downstream of the target site.

Our results show that the tool significantly increases cytidine deaminations by ~350-fold within a target region of up to 55 kb and introduces an average of 0.3 cytidine deamination per kb with a mutation occurring every 1–1.5 kb. The extended reach and targetability enable the optimization of entire heterologous pathways in *S. cerevisiae*, as we have demonstrated through increasing the efficiency of lycopene production in yeast by two-fold after only a single round of mutagenesis. CoMuTER bridges the current technological gap between genome-wide and narrowly localized mutagenesis and proves an efficient tool for the optimization of complex biosynthetic pathways.

## Results

### Design of different Cas3-base editor fusion constructs

The core of our random mutagenesis system is the class 1 type I-E CRISPR-Cas system native to *E. coli*, in particular the helicase/nuclease Cas3. The native interference process of this CRISPR-Cas system is illustrated in Fig. 1a. The first step is the recognition of a target site by the surveillance complex comprising the Cascade complex and a crRNA. The Cascade complex is composed of five proteins (Cas5, Cas6, Cas7, Cas8, and Cas11) with different stoichiometry that assembles around a 61-nucleotide crRNA with a 32 bp spacer sequence homologous to the target site[37,38]. crRNA maturation is accomplished by Cas6 which recognizes, binds to, and cuts a stem-loop at the 3′-end of a pre-crRNA to generate the mature crRNA. Six Cas7 subunits then assemble along the spacer sequence to form the backbone of the Cascade complex. Cas5 binds the 5′-end of the crRNA while Cas8 recognizes the protospacer-adjacent motif

(PAM) and mediates Cas3 recruitment. Two copies of Cas11 form the belly of the Cascade complex that stabilizes the crRNA and targets DNA. After the invading DNA has been recognized by the surveillance complex, a stable R loop is formed between crRNA and target DNA causing a conformational change in the Cascade complex, which in turn leads to the recruitment of Cas3. Cas3 nicks the non-target strand and loads it into its helicase domain. The non-target strand is subsequently reeled and unwound in 3′–5′ direction, thereby generating large stretches of ssDNA. The resulting ssDNA is intermittently cut by Cas3 leading to degradation of the invader DNA[39,40].

The ability to generate long stretches of ssDNA at targetable locations makes Cas3 a promising fusion partner for cytidine deaminases. These DNA-modifying enzymes can act on single-stranded DNA substrates. The fusion with a targetable helicase allows to generate a base editor with an action radius that is wide enough to encompass complex metabolic pathways (Fig. 1b, c).

The base editing system that we developed consists of two main components: (i) a fusion protein containing Cas3, a cytidine deaminase, and a uracil-DNA glycosylase inhibitor, referred to as 'Cas3-base editors' and (ii) the Cascade complex that processes and binds the crRNA (Fig. 1c). All genes encoding subunits of the Cascade complex were codon optimized and integrated into different loci in the *S. cerevisiae* (S288c) genome (Supplementary Table 5). The expression of each subunit was controlled by native *S. cerevisiae* promoters and terminators (Supplementary Data 1, Cascade subunits). The resulting strain was termed S288c-Cascade-background (S288c-CB).

In its native context, Cas3 nicks and degrades the non-target strand via its nuclease domain (Fig. 1a). Because DNA degradation could be detrimental to the cell, we investigated the requirement of the nuclease activity for the proper functioning of the CoMuTER system. To this end, we assessed both, a fully functional Cas3 and a nuclease-deficient Cas3 (dnCas3, containing H74A and D75A[41,42]) for the Cas3-base editor fusion constructs. Moreover, we tested two cytidine deaminases—*Pm*CDA1 from *Petromyzon marinus* (sea lamprey) and *r*APOBEC1 from *Rattus norvegicus* (rat). Both have already been shown to perform efficient site-directed mutagenesis when fused to an endonuclease-deficient Cas9 (dCas9)[43,44]. We decided to use the same fusion architectures as these previous studies, with *Pm*CDA1 fused C-terminally to Cas3, while *r*APOBEC1 was fused to the N-terminus of Cas3 (Fig. 1b). To avoid base-excision repair upon cytidine deamination, and thus the restoration of the native sequence, we also included the uracil-DNA glycosylase inhibitor (UGI) from bacteriophage PBS2 in the fusion construct[45]. Each of these three components (Cas3, cytidine deaminase, and UGI) was connected by specific linker peptides[46] that allow for sufficient flexibility to avoid steric interference while at the same time preventing the generation of coiled peptide structures. In total, this resulted in four different Cas3-base editor fusion constructs: Cas3-APOBEC1, Cas3-CDA1, dnCas3-APOBEC1, and dnCas3-CDA1 (Fig. 1b and Supplementary Data 1, Cas3-fusion constructs). In addition, we also included the unfused wild-type Cas3 in our study to compare the mutagenic effect of the native system to the Cas3-base editors. Each Cas3-base editor, as well as the unfused Cas3, was codon optimized and encoded on a yeast centromeric plasmid (Supplementary Data 1, Cas3-base editor plasmids). The expression of the Cas3-base editor/unfused Cas3 was controlled by a galactose-inducible promoter (pGAL1–10). This plasmid also encoded the crRNA cassette comprising a 32 bp spacer sequence flanked by the native *E. coli* CRISPR array repeat sequences and preceded by the native leader sequence. Expression in *S. cerevisiae* was enabled by the polymerase III (Pol III) promoter *SNR52* (Fig. 1b). Plasmids containing crRNA and Cas3-base editors or unfused Cas3 were transformed into S288c-CB. The resulting strains were used to study the system's functionality.

## Cas3-base editors are inducible and targetable

To test the activity of CoMuTER, we targeted the different fusion constructs and unfused Cas3 to the *CAN1* marker gene. Cells harboring an inactive *CAN1* allele are able to grow in the presence of the otherwise toxic compound canavanine, allowing for the selection of various loss-of-function mutations in *CAN1*[47]. Therefore, the number of cells in a population that can grow on selective canavanine-containing medium upon induction of the Cas3 base-editing system versus spontaneous mutants in a population where the system was not present or

induced offers a good proxy for the mutagenic activity of the system. A downside of this approach is that mutations that do not cause a loss of the Can1 function cannot be selected (biased selection). Therefore, this approach does not allow quantifying the exact efficiency of the different fusion constructs, but it does offer a quick way of assessing the mutagenic activity of the different Cas3-base editors.

We targeted each Cas3-base editor fusion construct as well as unfused Cas3 to two sites, 675 bp upstream or 68 bp downstream of the *CAN1* start codon (Fig. 2b and Supplementary Table 7). These

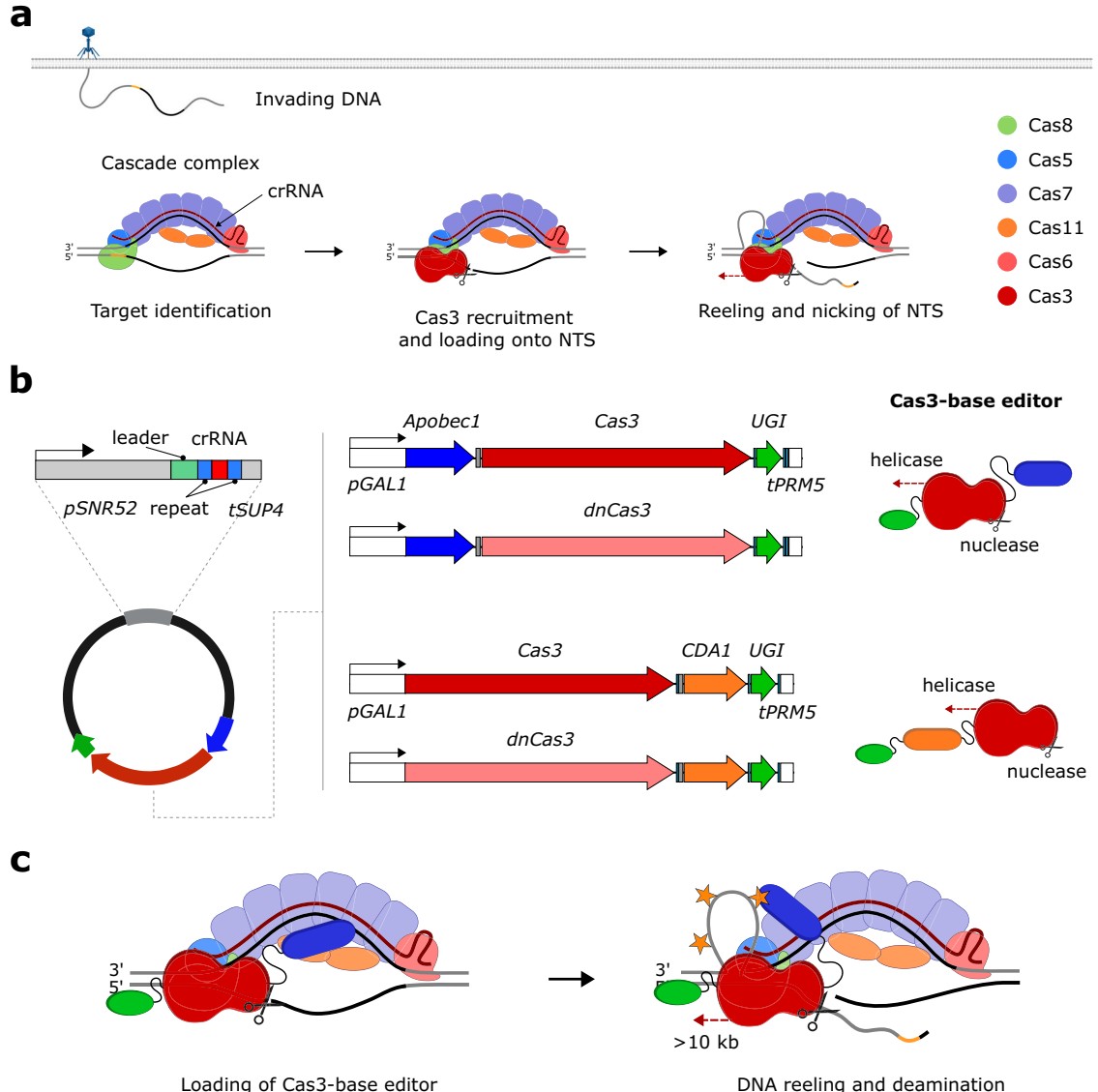

**Fig. 1 | Principle and design of CoMuTER. a** The native interference process of the type I-E CRISPR–Cas system in *E.coli* K-12. The Cascade complex, composed of Cas6 (pink), six copies of Cas7 (purple), Cas5 (blue), Cas8 (green), and two copies of Cas11, assembles around a mature crRNA (dark red) to recognize invading DNA. After the target site (black) has been recognized, conformational changes in the Cascade complex lead to the recruitment of Cas3 (red). Cas3 nicks the non-target strand (NTS) and loads it into its helicase domain. Subsequent reeling and unwinding of the non-target strand in the 3′–5′ direction generate large stretches of ssDNA. Intermittent cutting of the resulting ssDNA by Cas3 leads to the degradation of the invader DNA (see main text for details). Protospacer-adjacent motif is shown in yellow. **b** Design of Cas3-base editor fusion constructs and crRNA expression cassette. *Left* Fusion constructs and crRNA were encoded on a yeast centromeric plasmid. Expression of the crRNA is controlled by the *S. cerevisiae SNR52* promoter and *SUP4* terminator. The 32 nt spacer sequence (red) is flanked

by the native *E. coli* CRISPR array repeat sequences (blue) and preceded by the native leader sequence (green). *Middle* Cas-base editors comprise a cytidine deaminase (rAPOBEC1 or *pm*CDA1), Cas3 or nuclease-deficient Cas3 (dnCas3), and a uracil-DNA glycosylase inhibitor (UGI). Expression of Cas3-base editors was controlled by a galactose-inducible promoter (p*GAL1*) and a strong endogenous terminator *(tPRM5)*. Fusion constructs have a length of 4466 and 4382 bp for Cas3-APOBEC1/dnCas3-APOBEC1 and Cas3-CDA1/dnCas3-CDA1 fusion constructs, respectively. *Right* Illustration of the Cas3-base editors after translation. See the text for further information on fusion constructs. **c** Schematic of CoMuTER strategy exemplifying the Cas3-APOBEC1 base editor. Target site identification by Cascade leads to the recruitment of the Cas3-base editor. APOBEC1 (blue) introduces cytidine deaminations (stars) in the ssDNA generated by Cas3 (red) while base-excision repair is inhibited by UGI (green).

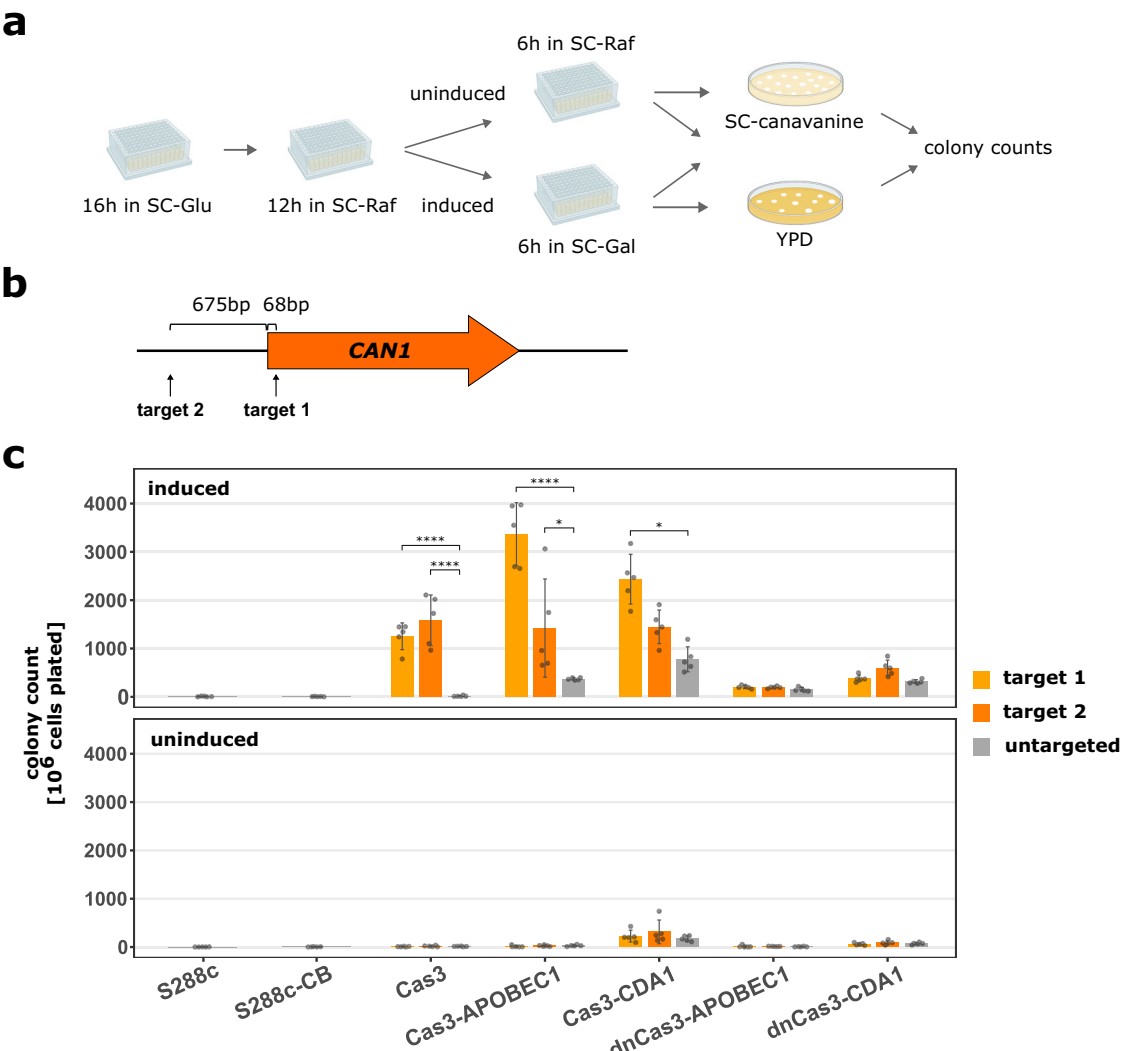

**Fig. 2 | Induction of CoMuTER causes a significant increase in *CAN1* knock-out mutations upon targeting.** The Cas3-base editor fusion constructs were targeted to the *CAN1* marker gene to assess their inducibility and activity. **a** Workflow of CoMuTER induction and activity assessment. A detailed description of all steps can be found in the "Methods" section. SC-Glu, SC medium with 2% glucose; SC-Raf, SC medium with 2% raffinose; SC-Gal, SC medium with 2% galactose and 2% raffinose. **b** Target sites are located 68 bp downstream and 675 bp upstream of the *CAN1* start codon, respectively. Sites are located directly downstream of a 5′-AAG-3′ PAM. **c** Average (bars) and individual number (points) of colonies that grew on selective medium for strains expressing the four Cas3-base editors, unfused Cas3, and two control strains (S288c and S288c-CB) targeted to either target site 1 or 2, or untargeted. Untargeted base-editors lack the spacer (target) sequence in the crRNA expression cassette but instead contain a 20 bp "placeholder" DNA which is not present in the *S. cerevisiae* genome (see the "Methods" section, crRNA cassette).

Strains S288c and S288c-CB are always untargeted, as they do not contain a crRNA cassette. To assess the mutagenic activity of each Cas3-base editor the number of colonies per strain was compared to the respective untargeted version. Data are presented as mean ± SD of $n = 5$ independent biological replicates per strain and the target site. Data were transformed to proportions of cells containing a knock-out mutation and analyzed using a generalized linear mixed-effects model with a binomial link function. To test for significant differences per strain between targeting site 1/2 and not targeting we used a two-sided post-hoc test with Sidak adjustment, $*p \le 0.05$, $****p \le 0.0001$. Specifically, for targets 1 and 2, respectively, $p$-values are $6.66 \times 10^{-7}$ and $2.22 \times 10^{-16}$ for Cas3, $1.5 \times 10^{-5}$ and 0.018 for Cas3-APOBEC1, 0.043 and 0.448 for Cas3-CDA1, 0.161 and 0.596 for dnCas3-APOBEC1, 0.64 and 0.336 for dnCas3-CDA1. All averages can be found in Supplementary Table 1. Source data for this figure are provided as a Source Data file.

target sites are located directly downstream of a 5′-AAG-3′ PAM as previously reported in *E. coli*[33,41,48]. We grew each strain in a synthetic complete medium containing 2% galactose to induce the expression of the CoMuTER system and plated it on a canavanine-containing medium (see the "Methods" section). The strains' activities were examined against their uninduced counterparts (Fig. 2a). To assess the mutagenic activity of each Cas3-base editor/unfused Cas3, the resulting number of colonies was compared to the respective untargeted version (Fig. 2c). Only Cas3-APOBEC1, Cas3-CDA1 and unfused Cas3 (systems with an active Cas3 nuclease domain) caused a significant increase in the number of colonies upon targeting compared to their untargeted counterpart which lacked the 32 bp spacer sequence in the

crRNA expression cassette. We observed a general increase in the number of colonies for all strains featuring a cytidine deaminase, independent of targeting, compared to the parental strains S288c and S288c-CB (see the "Discussion" section). However, this activity is significantly lower than the target site-specific activity of the Cas3-base editors featuring Cas3 with an active nuclease domain. Strains in which the expression of the CoMuTER system was not induced did not show any significant increase in colony number. The slightly increased number of colonies in strains expressing Cas3-CDA1 under uninduced conditions is possibly caused by a combination of leaky expression from the *GAL1-10* promoter[49,50] driving the expression of Cas3-CDA1, and the highly active *pm*CDA1 cytidine deaminase[20,42].

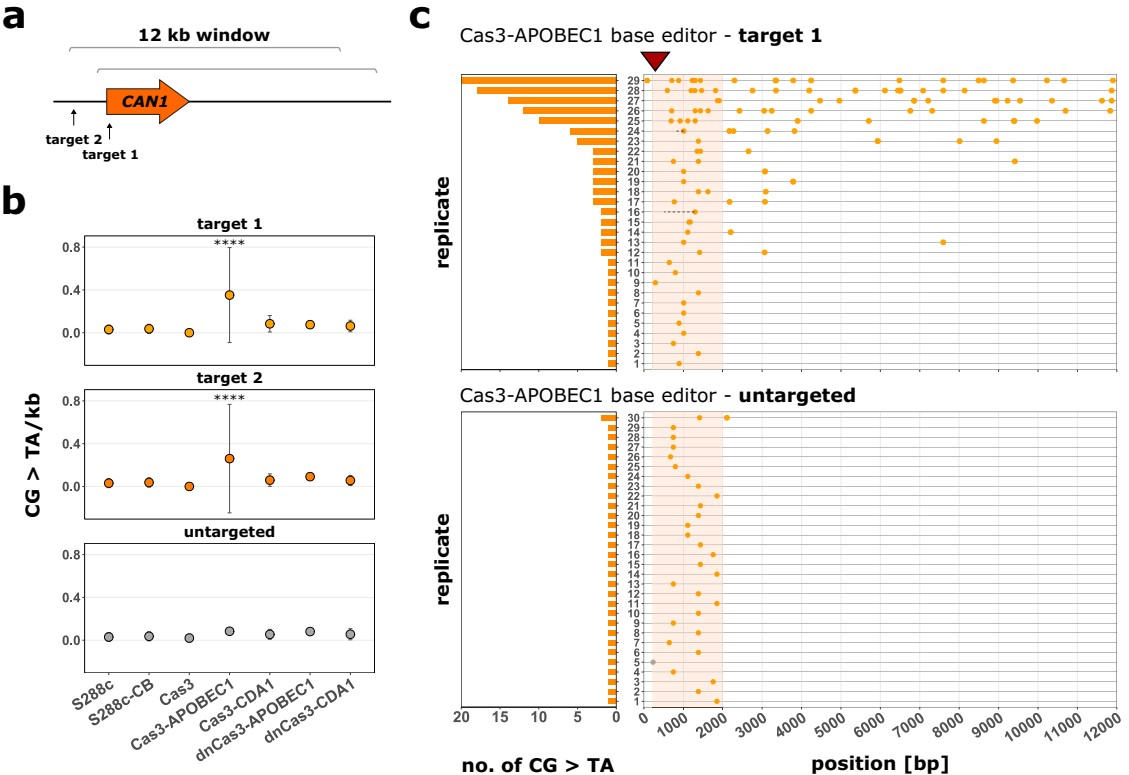

**Fig. 3 | Cas3-APOBEC1 base editor introduces random cytidine deaminations in a 12 kb region downstream of the target site. a** The sequenced 12 kb windows downstream of target sites 1 and 2, respectively. **b** Average number of cytidine deaminations per kb within a 12 kb window downstream of the respective target site for strains expressing the four Cas3-base editors, unfused Cas3, and two control strains (S288c and S288c-CB) targeted to either target site 1 (yellow), 2 (orange), or untargeted (gray). Strains S288c and S288c-CB are always untargeted, as they do not contain a crRNA cassette. For easier comparison, their respective averages are included in the graphs for target sites 1 and 2. Data are presented as mean ± SD of $n$ = at least 21 independent biological replicates per strain and target site (see Supplementary Table 8 for an exact number of replicates per strain and target site). Data were fitted using a generalized linear mixed-effects model. A two-sided post-hoc test with Sidak adjustment was used to identify strains with significant differences to the control strain (S288c), ****$p < 0.0001$. Specifically, $p$-values are $3.11 \times 10^{-5}$ and $5.45 \times 10^{-8}$ for Cas3-APOBEC1 targeted to target sites 1 and 2, respectively, and >0.75 for all other samples (exact $p$-values can be found in Supplementary Data 1, $p$-values). **c** Position and a number of cytidine deaminations (orange circles) introduced within a 12 kb window in each biological replicate of strains expressing Cas3-APOBEC1 targeted to target site 1 (upper panel) or untargeted (lower panel). Other SNVs are indicated by gray circles and deletions by a dashed line. Red triangle indicates target site 1 and shaded orange boxes indicate the position of the *CAN1* marker gene. Source data for this figure are provided as a Source Data file.

These initial experiments indicated that the Cas3-base editing system is inducible and active in *S. cerevisiae* and can be targeted to specifiable target sites. Moreover, the nuclease activity of Cas3 proved vital for the target site-specific activity of the respective base editor.

## Cas3-APOBEC1 inserts random cytidine deaminations in target region

To examine the mutational spectrum of the Cas3-base editors, we sequenced a 12 kb region downstream of the two previously targeted sites located 68 bp downstream and 675 bp upstream of the *CAN1* start codon, respectively (Fig. 3a). Similar to the previous experiment, we selected cells based on their ability to grow on canavanine-containing medium after induction of the CoMuTER system. We analyzed 30 replicates per strain and per target site as well as untargeted strains (Supplementary Figs. 1–7).

The results revealed that only the Cas3-APOBEC1 base editor caused a significant increase in cytidine deaminations in the sequenced region compared to the parental strains S288c and S288c-CB. The base editor introduced an average of 0.35/0.26 cytidine deaminations per kb with a mutation occurring every 1000/736 bp for target sites 1 and 2, respectively (Fig. 3b, c and Supplementary Fig. 1). On average, the first observed cytidine deamination was introduced 731/766 bp downstream of target site 1/2. This increase was only observed when the Cas3-APOBEC1 base editor was targeted to the

*CAN1* locus, while almost all strains expressing the untargeted Cas3-APOBEC1 base editors contained only a single mutation in the sequenced region (Fig. 3b, d). The introduced mutations spanned the entire sequenced region, suggesting an activity window of at least 12 kb.

The vast majority of detected mutations in strains expressing Cas3-APOBEC1 were cytidine deaminations with 98.4/81% for target sites 1 and 2, respectively (Supplementary Fig. 8). However, 1.6/15% (target sites 1 and 2, respectively) of detected mutations were deletions, which are a likely consequence of the Cas3 nuclease activity (Supplementary Fig. 8). Base editors containing Cas3 with an inactive nuclease domain did not cause an increased number of cytidine deaminations within the sequenced window. This observation agrees with the previous data (Fig. 2c). Interestingly, Cas3-CDA1 did not significantly increase the number of cytidine deaminations in the targeted region (see the "Discussion" section), whereas unfused Cas3 introduced deletions downstream of the target site (Fig. 3b and Supplementary Figs. 2–5). When Cas3 was not targeted we did not observe any deletions, but instead found different types of point mutations in *CAN1*, as expected due to the selection of a canavanine-containing medium (Supplementary Fig. 5c). This confirms that the Cas3 base-editing system is targetable.

To obtain a better view of the target window and examine the occurrence of off-target mutations, we sequenced the complete

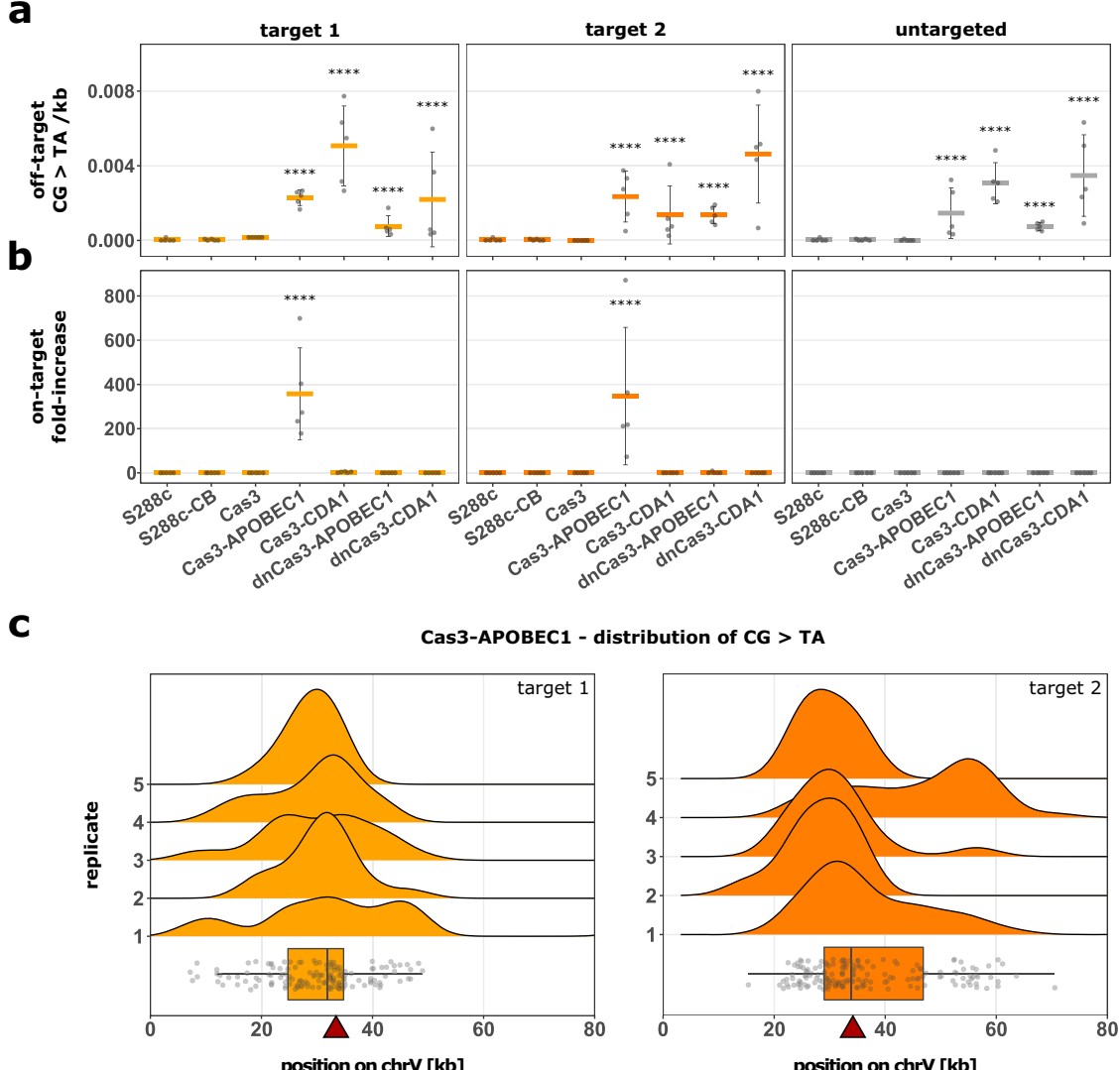

**Fig. 4 | Bi-directional activity of Cas3-APOBEC1 increases the number of cytidine deaminations in the targeted region 350-fold.** Number of cytidine deaminations in the target region (on-target) or in the rest of the genome (off-target) is determined by whole genome sequencing of strains expressing one of the four Cas3-base editors, unfused Cas3 and the two control strains (S288c and S288c-CB) targeted to either target site 1 (yellow), 2 (orange), or untargeted (gray). **a** Average number of off-target cytidine deaminations per kb for each strain and target site. Strains S288c and S288c-CB are always untargeted, as they do not contain a crRNA cassette. For easier comparison, their respective averages are included in the graphs for target sites 1 and 2. Data are presented as mean (horizontal lines) ± SD of $n = 5$ independent biological replicates (points) per strain and target site. Data were fitted using a generalized linear mixed-effects model. A two-sided post-hoc test with Sidak adjustment was used to identify strains with significant differences to the control strain (S288c), ****$p \leq 0.0001$. Specifically, $p$-values are 1, 0.205, $1.73 \times 10^{-8}$, $7.99 \times 10^{-12}$, $8.40 \times 10^{-5}$, $2.40 \times 10^{-8}$, 1, 1, $1.34 \times 10^{-8}$, $1.25 \times 10^{-6}$, $1.25 \times 10^{-6}$, $2.07 \times 10^{-11}$, 1, 0.993, $7.14 \times 10^{-7}$, $1.12 \times 10^{-9}$, $9.74 \times 10^{-5}$ and $3.51 \times 10^{-10}$ (from left to right). **b** Fold-increase in cytidine deaminations in the targeted region compared to the rest of the genome (target region defined as a region on chromosome 5 in which 99.3% of cytidine deaminations were observed in strains expressing Cas3-APOBEC1). Data and statistical analysis are similar to (**a**). The $p$-values are $5.85 \times 10^{-9}$ for Cas3-APOBEC1 target site 1, $1.40 \times 10^{-8}$ for Cas3-APOBEC1 target site 2, and 1 for all other samples. **c** Distribution of cytidine deaminations around target site 1 (left panel) and 2 (right panel) in $n = 5$ independent biological replicates of strains expressing Cas3-APOBEC1. Boxplots represent the distribution of all deaminations (gray points) found in the 5 biologically independent replicates shown above. The top, middle line, and bottom of the box represent the upper quartile (Q3), median, and lower quartile (Q1), respectively. Tukey-style whiskers extend to a minimum and maximum of 1.5× interquartile range beyond the box. Red triangles indicate the respective target site. *X*-axis was cut at 80 kb. Source data for this figure are provided as a Source Data file.

genome of a subset of replicates (5 replicates/strain/target site; resulting in a total of 85 sequenced lineages). We did not observe an increase in off-target deletions in any of the analyzed strains (Supplementary Fig. 9). However, all Cas3-base editors cause a significant increase in cytidine deaminations across the whole genome, independent of targeting (Fig. 4a). This is likely caused by the presence of the respective cytidine deaminase in each of these strains. However, in cells expressing the Cas3-APOBEC1 base editor the number of cytidine deaminations was on average increased by 347/357-fold (target sites 1 and 2, respectively) in the targeted region compared to the rest of the

genome (Fig. 4b). This increase was only observed when Cas3-APOBEC1 was targeted to *CAN1*, again demonstrating efficient targeting.

In addition to investigating the off-target activity of the Cas3-base editors, whole genome sequencing allowed the assessment of the action radius of the Cas3-APOBEC1 base editor. The activity window, defined as the region in which 99.3% of cytidine deaminations were observed, reaches around 19/21.5 kb downstream of the target sites 1 and 2, respectively (Fig. 4c). Surprisingly, we also observed an increased number of cytidine deaminations upstream of the respective

target sites. This bi-directional activity has been reported for type I-A[51], type I-C[34], and type I-D[52, 53], but to our knowledge not for type I-E CRISPR-Cas systems. The activity window reached up to -15.5/36 kb upstream of target sites 1 and 2, respectively (Fig. 4c). Based on these observations, the Cas3-APOBEC1 base editor has a total activity window of around 37/55 kb, with the mutation frequency gradually tapering off near the edges of the region. There was no systematic distribution pattern of mutations in-between biological replicates, indicating that cytidine deaminations are introduced randomly, as expected.

## CoMuTER allows optimizing a heterologous biosynthesis pathway

The extended reach and capacity to introduce random cytidine deaminations within definable genomic regions make the Cas3-APOBEC1 base editor a promising tool for heterologous pathway optimization. We tested this application of the Cas3-base editor by targeting a heterologous lycopene synthesis pathway that was inserted into the *S. cerevisiae* genome. Lycopene is a carotenoid with high commercial value due to its antioxidant, anticancer, and anti-inflammatory properties[54,55]. Moreover, its bright red color allows easy selection of colonies with improved lycopene production since the color intensity is directly correlated with lycopene synthesis.

For this proof-of-concept, *S. cerevisiae* strain CEN.PK2-1C was chosen, as it is a standard strain for metabolic engineering and is used in several industrial processes[56–58]. After integration of the five genes required for cascade complex formation, we inserted the three heterologous lycopene biosynthesis genes[59], geranylgeranyl diphosphate synthase (*CrtE*, from *X. dendrorhous*), phytoene synthase (*CrtB*, from *P. agglomerans*), and phytoene desaturase (*CrtI*, from *X. dendrorhous*), as well as truncated HMG-CoA reductase (*tHMG1*, from *S. cerevisiae*) (Fig. 5a, b). These heterologous genes are controlled by yeast-endogenous promoters and terminators and separated from each other by 60 bp spacer sequences (Supplementary Data 1, Lycopene cassette). The resulting ~9 kb-cassette was introduced 436 bp upstream of the native *CAN1* gene to enable parallel selection of cells with an active CoMuTER system. The resulting lycopene base strain yielded orange-colored colonies, indicating functional expression of the introduced pathway (Fig. 5e, bottom right colony). Lastly, the plasmid-encoded Cas3-base editor targeted to a site directly upstream of the heterologous cassette was introduced (Supplementary Table 7).

After induction of the system, cells were plated on a canavanine-containing medium and screened for colonies with increased lycopene production. We observed colonies that ranged in color from white, suggesting impaired or no lycopene production, to more intense red, which suggests increased lycopene production. We selected six colonies that showed a more intense red color compared to the non-mutagenized base strain (Fig. 5c). Sequencing the lycopene pathway of the selected colonies revealed that each colony contained between 1 and 8 cytidine deaminations introduced throughout the pathway, occurring in the promoter, terminator and coding regions (Fig. 5d). Cytidine deaminations in the coding regions resulted in both synonymous and non-synonymous mutations. Each sequenced strain contained at least one amino acid substitution in the lycopene pathway caused by cytidine deamination. Interestingly, 2 of the amino acid substitutions found in *crtB* were present in more than one strain, S228F (strains 4, 5, and 6) and M244I (strains 1 and 2).

To assess whether the identified mutations cause the observed change in phenotype, each mutated pathway was amplified by PCR and used to re-create the lycopene base strain. Indeed, the resulting strains maintained the enhanced coloration when compared to the non-mutated base strain (Supplementary Fig. 11 and Fig. 5e). Next, the selected strains were grown in liquid culture for lycopene extraction followed by LC–MS to quantify lycopene levels and examine the correlation between lycopene content and color. Five out of the six strains

show a significant increase in lycopene production with more than 2-fold increases for the best-producing strain compared to the original non-mutagenized strain (Fig. 5e). To determine whether the introduced mutations affect gene expression levels in the selected strains, relative expression of the lycopene pathway genes was determined via quantitative real-time PCR (qPCR). None of the selected strains showed a significant difference in relative mRNA expression of the examined genes compared to the base strain (Supplementary Fig. 12). Strikingly, the three best-performing strains, strains 4, 5, and 6, contained the same mutation in the *crtB* gene causing a serine to phenylalanine substitution at position 228. The differences in lycopene production between these strains are not significant (more details in Supplementary Data 1, *p*-values). Moreover, strain 5 did not contain any additional mutation. This suggests that crtB S228F is the sole driver of the increased lycopene production in the three best-performing strains.

To test the activity of CoMuTER at a different genomic location, we targeted a site 386 bp downstream of the essential gene *SEC14*. *SEC14* encodes a phosphatidylinositol transfer protein which is essential for intracellular lipid metabolism[60]. Importantly, Sec14 is the only target of a class of small molecule inhibitors termed nitrophenyl(4-(2-methoxyphenyl)piperazin-1-yl)methanones (NPPMs), used to inhibit the growth of pathogenic fungi. Several mutations in *SEC14* have been reported to confer resistance to the NPPM 481, while maintaining Sec14 function[12,61,62], making it an interesting target to test the ability of the CoMuTER system to generate resistant variants.

After induction of the CoMuTER system, cells were plated on a medium containing 3 μM NPPM 481 to select for variants containing resistance-conferring mutations (see the "Methods" section for further information about the chosen concentration and experimental setup). We found an average of 10 colonies per plate (corresponding to 0.02% of plated cells) that were able to grow in the presence of the small-molecule inhibitor (Supplementary Table 2). Control strains expressing either no Cas3-base editor (CEN.PK-CB) or an untargeted Cas3-base editor (Cas3-APOBEC1, untargeted) showed an average of -0.22 and -1.66 colonies per plate, respectively (corresponding to 0.00044% and 0.00332% of plated cells, respectively). We subsequently sequenced the *SEC14* locus (171 bp upstream to 597 bp downstream of the start codon) of 18 colonies that showed resistance to NPPM 481 after CoMuTER activity. Individual colonies contained between 1 and 7 cytidine deaminations with an average of -2 cytidine deaminations per colony (Supplementary Table 3). Importantly, all identified mutations (37 in total) were cytidine deaminations. Among these, 18 caused amino acid substitutions, resulting in seven unique amino acid changes: H112Y, E150K, V154I, S173L, S183F, G210S, and S222F. The remaining 19 mutations caused either synonymous changes or were outside the CDS (Supplementary Table 3). Of the seven unique amino acid substitutions, residues H112, V154, S173, and G210 have been previously reported to confer resistance to NPPM 481[12,62,63] further strengthening our screening results. These data demonstrate the capabilities of CoMuTER in a different genomic context and its ability to introduce resistance-conferring mutations in the essential gene *SEC14*.

## Discussion
Random mutagenesis is a valuable tool to optimize specific phenotypes, including the production of specific compounds by heterologous metabolic pathways. Traditional random mutagenesis targets the complete genome, which carries the inherent risk of generating undesirable mutations on top of the desirable mutations in specific targets. Conversely, strategies to introduce random mutations in a targeted region are often limited by relatively small activity windows (<3 kb) that are insufficient to diversify multiple genes or entire pathways. CoMuTER fills this gap by providing a targetable and inducible tool capable of introducing random cytidine deaminations across large genomic regions of up to 55 kb.

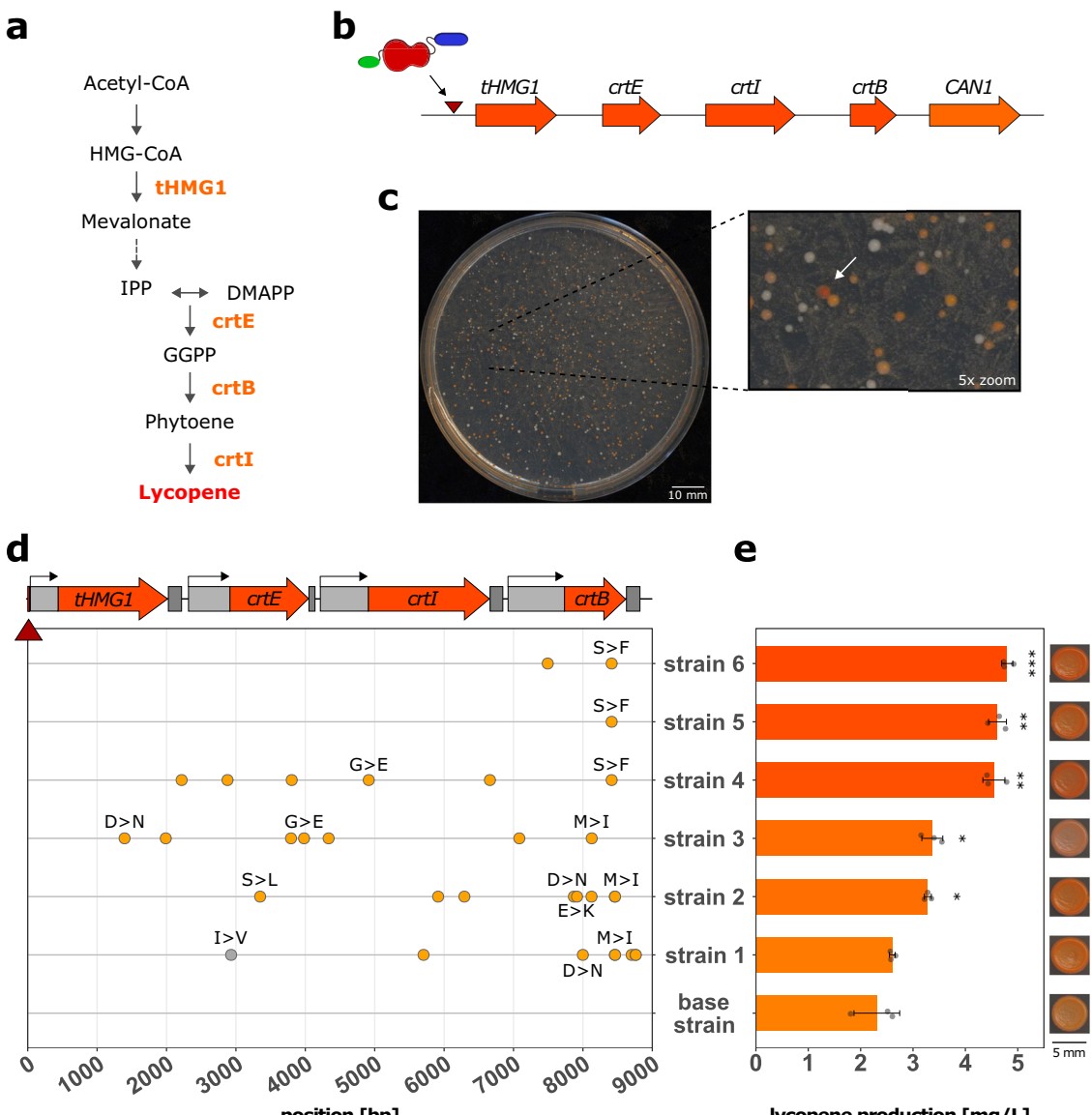

**Fig. 5 | CoMuTER optimizes lycopene production after a single round of mutagenesis.** The Cas3-APOBEC1 base editor was tested for its capability to improve the performance of a heterologous pathway. **a** Simplified schematic of the engineered lycopene biosynthetic pathway in *S. cerevisiae*. Integrated enzymes are shown in orange. The dashed arrow signifies three intermediate steps native to *S. cerevisiae* (not shown). Enzyme abbreviations: tHMG1 truncated 3-hydroxy-3-methylglutaryl-CoA reductase, CrtE geranylgeranyl diphosphate synthase, CrtB phytoene synthase, CrtI phytoene desaturase. **b** The four lycopene biosynthesis genes (orange) were integrated 436 bp upstream of the *CAN1* gene (see main text for details). The total length of the introduced pathway was 8792 bp. The Cas3-base editor was targeted directly upstream of the promoter driving *tHMG1* expression (red triangle). **c** Example of variation in colony color and size after one round of CoMuTER mutagenesis. The arrow shows one of the more intensely colored colonies. Scale bar represents 10 mm. **d** Position of mutations introduced in the integrated lycopene biosynthesis genes (top) of six selected strains. Each selected strain contained between 1 and 8 cytidine deaminations (orange circle). Other SNVs are represented by a gray circle. Amino acid substitutions caused by introduced mutations are denoted in single letter code. The target site is indicated by a red triangle. **e** Lycopene production of six selected strains compared to the non-mutagenized base strain. Corresponding phenotypes are shown on the right (uncropped image in Supplementary Fig. 11). Scale bar represents 5 mm. Data are presented as mean (bars) ± SD of $n = 3$ independent biological replicates (points). Significant difference to base strain was determined with unpaired two-sided two-sample *t*-test, df = 4, $*p \leq 0.05$, $**p \leq 0.01$, $***p \leq 0.001$. Specifically, p-values are $7.43 \times 10^{-7}$, 0.0014, 0.0016, 0.0194, 0.0176, and 0.4424 (from top to bottom). Source data for this figure are provided as a Source Data file.

The core of CoMuTER is the class 1 type I CRISPR-Cas system. Class 1 systems have recently been used for genome editing and gene regulation in both bacteria and eukaryotes[34–36,51–53,64–67]. Here, we repurposed the type I-E CRISPR-Cas system for targeted in vivo mutagenesis by capitalizing on the helicase activity and targetability of Cas3 in fusion with a cytidine deaminase. We tested two different cytidine deaminases, *pm*CDA1 and *r*APOBEC1, because their activity is well documented and has been employed in other mutagenic tools[41,42]. Interestingly, only Cas3-APOBEC1 caused a significant increase in cytidine deaminations in the targeted region, while Cas3-CDA1 only

lead to a minor increase in mutation frequency (Fig. 3b, c and Supplementary Fig. 2). This could be a result of the fusion direction of the Cas3-CDA1 base editor. In contrast to APOBEC1, which was N-terminally fused to Cas3, CDA1 was fused to the C-terminus of Cas3 (Fig. 1, orientation based on reported fusion with dCas9[41,42]). This orientation might sterically hinder the helicase activity of Cas3, while still allowing CDA1 to be active in the region proximal to the targeted site. Cas3-CDA1 is recruited to the target site where CDA1 can introduce cytidine deaminations but sterically prevents DNA unwinding. In the Cas3-APOBEC1 base editor, Cas3 was C-terminally fused to the

smaller UGI (84 amino acids), potentially permitting the reeling of DNA and thereby result in the extended reach of the base editor and increased frequency of cytidine deaminations.

The inherent threat to cellular integrity posed by cutting DNA led us to explore the importance of Cas3's nuclease domain. Our results showed that the CoMuTER system is dependent on the nuclease activity of Cas3 since nuclease-deficient Cas3 variants did not show target-site-specific activity. Cryo-EM analysis of the *Thermobifida fusca* type I-E system suggests that the initial nicking of the non-target strand enables its threading into the helicase domain and initiates the next phase of processive DNA unwinding[68–70]. These findings explain the inactivity of nuclease-deficient Cas3-base editors. Nevertheless, the nuclease activity of Cas3 may lead to occasional deletion events in addition to the deaminations generated by the cytidine deaminase. Indeed, some replicates selected after induction of CoMuTER contained deletions (Fig. 3c, Supplementary Figs. 1 and 2). For the Cas3-APOBEC1 base editor targeted to site 1 (Fig. 3c), the number of deletion events is relatively low (1.6%) compared to the number of cytidine deaminations (98.4%). Interestingly, the occurrence of deletions and general mutational outcomes vary between the two tested target sites, with the second target site showing 15% deletion events in addition to the expected point mutations (81%). The reason for this difference remains obscure, although we hypothesize that the local chromatin structure around the respective target sites may contribute to the effect.

A primary feature of CoMuTER is its targetability to a specified locus in the genome. We observed that the Cas3-base editor is only active upon targeting (Figs. 3 and 4), but strains expressing a cytidine deaminase fused to Cas3 or dnCas3 showed a significant increase in the number of off-target cytidine deaminations compared to the parental S288c strain (Fig. 4a). This increase is independent of the crRNA-guided targeting of the Cas3-base editor. Guide RNA-independent off-target mutagenesis has already been reported for cytosine base editors and is likely caused by the intrinsic DNA affinity of the deaminase domain[71,72]. Importantly, however, this off-target activity is marginal compared to the target site-specific activity of the Cas3-APOBEC1 base editor, which is on average ~350 times greater. Therefore, CoMuTER provides an advantage over genome-wide mutagenesis by limiting the risk of off-target mutations.

Interestingly, the Cas3-APOBEC1 base editor showed bi-directional activity of up to ~21.5/19 kb downstream and ~15.5/36 kb upstream of the targeted sites, depending on the target site (Fig. 4c). This observation was unexpected since Cas3 is reported to have a unidirectional 3′–5′ helicase activity[33,35,36,64]. However, bi-directional activity has been described for type I-A[51], -C[34], and -D[52,53] CRISPR-Cas systems. Redding et al. showed that under specific conditions type I-E Cas3 can translocate in either direction from the Cascade-bound DNA target site in the presence of the Cas1–Cas2 adaptation complex. In this instance, the nuclease activity of Cas3 is attenuated which may prevent degradation of the targeted DNA[73]. Similarly, we observed a low percentage of replicates containing deletions within the Cas3 activity window. This could be a result of the fusion of *r*APOBEC1 to the N-terminal nuclease domain of Cas3, which may impede DNA cleavage.

A key consideration in the development of any mutagenic instrument is its ultimate applicability. CoMuTER's 55 kb reach makes it a highly suitable tool for optimizing the synthesis of useful biomolecules using cell factories. Innumerable diverse compounds can be produced from heterologous pathways and often require duplication of basal metabolic genes to meet cellular demand. Altogether, these genetic additions add up to substantial pathway lengths and provide an excellent opportunity to test the abilities of our tool. Highly valuable compounds including lycopene (4 genes, ~9 kb, this study), resveratrol (3 genes, ~5-6 kb)[74], artemisinin (5 genes, ~6-7 kb)[75], or tetrahydrocannabinol (16 genes, ~32 kb)[76] have pathway lengths that escape the reach of existing tools for random mutagenesis but fall

comfortably in range for CoMuTER when combined into a singular locus. The lycopene pathway was chosen as a proof-of-concept, due to its exemplary length and easy screenable phenotype. Sequencing analysis revealed strong diversity in the number and position of introduced mutations in the pathway (Fig. 5d). Although we identified mutations in the promoter and terminator regions of the introduced lycopene pathway genes in the selected strains, relative gene expression levels were unchanged when analyzed by qPCR (Supplementary Fig. 12). This suggests that increases in lycopene production are either caused by alterations on a translational level or changes in the catalytic activity of *tHMG1*, *crtE*, *crtI*, and *crtB*. Indeed, we were able to identify a mutation in crtB, S228F, that caused a two-fold increase in lycopene production. To our knowledge, this amino acid substitution has not previously been identified.

A limitation of CoMuTER is that the cytidine deaminase generates C to T and G to A transitions and can therefore only accomplish ~9% of all possible amino acid substitutions. Even still, CoMuTER was able to dramatically improve the production of lycopene after a single round of induction and selection. An obvious complement to expand the mutational capacity of CoMuTER would be to generate a dual fusion construct featuring both a cytidine and adenine deaminase that act in tandem. This combination would increase the spectrum of possible amino acid substitutions to ~19%, which may further enhance the tool's potential. Another notable observation is that the mutation rate is not evenly distributed across the targeted region (Fig. 4c). However because CoMuTER allows in vivo mutagenesis, it is possible to easily generate a sufficiently large number of mutants to efficiently probe the targeted region, with the screening for superior variants being the main bottleneck.

The strengths of CoMuTER are derived from the exploits of the class 1 type I CRISPR-Cas system. The tool can be temporally induced and is targetable to any desired locus via a crRNA. In addition, Cas3 permits access to large genomic windows, like for example, an entire metabolic pathway. Generating ssDNA as a substrate for a cytidine deaminase avoids harmful double-strand breaks and makes CoMuTER a more tolerable tool for random mutagenesis. The high mutagenic activity accelerates the realization of improved phenotypes and provides non-obvious breakthroughs in as few as one round of application. Moreover, this can be achieved without a priori knowledge of the sites needing alteration. We demonstrated the capabilities of CoMuTER in two different genetic backgrounds in *S. cerevisiae*, but the system could be easily converted for use in other organisms, including higher eukaryotes, i.e. plants and mammalian cells where expression of Cas3 and a cytidine deaminase have already been demonstrated. Thus, CoMuTER can rapidly generate phenotypic diversity across species and provides a foundation to create efficient and sustainable production of high-value compounds in biotechnology applications.

## Methods

### Strains used in this study
Strains used in this work were derived from a prototrophic haploid S288c or auxotroph haploid CEN.PK2-1C. A full list of strains can be found in Supplementary Table 4.

### Design and construction of the CoMuTER system
Gene constructs encoding Cascade subunits as well as plasmids encoding Cas3 fusion constructs/unfused Cas3 and the crRNA cassette were ordered from BGI Genomics and verified by Sanger sequencing. Sequences and additional information about these gene constructs can be found in Supplementary Data 1.

### Cascade complex
The class 1 type I-E CRISPR-Cas Cascade complex consists of one copy of Cas5, Cas6, and Cas8, two copies of Cas11, and six copies of Cas7.

Genes encoding Cascade subunits (*Cas5, Cas5, Cas7, Cas8, Cas11*) were taken from *E. coli* K-12 MG1655 and sequenced optimized for expression in *S. cerevisiae* (IDT Codon Optimization Tool). For correct subcellular localization, two copies of the SV40 nuclear localization sequences (NLS) were introduced N- or C-terminally of each gene (depending on the availability of free N- or C-termini of each subunit in the Cascade complex). Gene expression was controlled by native *S. cerevisiae* promoters and terminators. For Cas7, Cas8, and Cas11 constitutive promoters with different strengths were chosen, while the expression of Cas5 and Cas6 was driven by the inducible *GAL1-10* promoter. Inducibility of Cas6 and Cas5 (responsible for crRNA processing and binding of the 5′ handle of the mature crRNA, respectively) prevents the formation of the Cascade complex under normal growth conditions. We employed the *GAL1-10* promoter for only two Cascade subunits to reduce competition for transcription factors and to lower the risk of recombination between similar promoter sequences. To account for the higher copy number of Cas7 in the assembled Cascade complex (six copies) we selected the strong *TDH3* promoter and *PRM9* terminator. Details and sequences of each gene construct can be found in Supplementary Data 1, Cascade subunits. Each Cascade gene construct was flanked by 500 bp sequences homologous to the respective integration site (Supplementary Table 5). Integration sites were chosen based on a study from Reider Apel et al.[77]. Gene constructs were amplified using PrimeSTAR GXL DNA polymerase (TaKaRa) and integrated into their respective integration sites via CRISPR-Cas9-assisted homologous recombination using pV1382[78]. A list of gRNAs targeting the different integration sites can be found in Supplementary Table 6. Yeast transformation was carried out using a DMSO-LiAc procedure as described in Pan, X et al.[79]. Primers used for integration and sequence verification were ordered from IDT; primer sequences are listed in Supplementary Data 1, Primers.

### Cas3-base editors

Cas3-base editors comprise Cas3 fused to a cytidine deaminase (*r*APOBEC1 or *pm*CDA1) and a uracil glycosylase inhibitor (UGI). The gene encoding Cas3 is taken from *E. coli* K-12 MG1655. The cytidine deaminase *Pm*CDA1 is taken from *Petromyzon marinus* (sea lamprey) and *r*APOBEC1 from *Rattus norvegicus* (rat). The gene encoding UGI is taken from PBS1 bacteriophage. All gene sequences were codon optimized for *S. cerevisiae* (IDT Codon Optimization Tool). The three components were connected by Xten linker peptides[44] and each fusion construct contained three NLS (SV40 or C-myc). The expression of the Cas3-base editors was controlled via the galactose-inducible *GAL1-10* promoter and the *PRM5* terminator, both native to *S. cerevisiae*. In addition to the Cas3-base editors containing an active nuclease domain, a set of fusion constructs featuring a nuclease-deficient Cas3 (dnCas3) were generated by introducing two amino acid substitutions in its HD domain (H74A and D75A[39,40]). The architecture and sequence of the four resulting Cas3-base editors as well as of the unfused Cas3 gene construct are detailed in Supplementary Data 1, Cas3-fusion constructs.

### crRNA cassette

To generate the crRNA expression cassette parts of the native *E. coli* CRISPR array sequence (directed repeats and leader sequence) were used. The native *E. coli* CRISPR array consists of directed repeats (29 bp) separated by spacers (32 bp) homologous to segments of viral genomes. The repeat-spacer array is preceded by an A-T-rich leader sequence required for correct spacer integration and array transcription[80]. The CRISPR array is transcribed into a precursor CRISPR RNA that is further processed to mature crRNA by the Cascade subunit Cas6. The processed crRNA consists of a 5 bp 5′ handle, a 32 bp spacer, and a 24 bp 3′ stem-loop (61 bp total length).

For the crRNA cassette, two repeats flanking one spacer sequence were used. The native 32 bp spacer sequence was replaced by a 20 bp

space holder (5′-GAGACGGAAGATTCCGTCTC-3′) containing two BsmBI restriction sites (type IIS restriction enzyme) that allow for flexible integration of the respective spacer/target sequence (Supplementary Table 7). This short array was preceded by a 54 bp segment of the native leader sequence. For expression in *S. cerevisiae* the *SNR52* promoter as well as two copies of the *SUP4* terminator were used. Details and sequences of each component of the crRNA cassette are listed in Supplementary Data 1, crRNA cassette.

The Cas3-base editors/unfused Cas3 as well as the crRNA cassette were encoded on the single copy plasmid pMV_hyg (Supplementary Data 1, Cas3-base editor plasmids). Plasmids were transformed into strains S288c-CB or CEN.PK-CB using a DMSO-LiAc procedure as described in Pan X. et al.[79]. and maintained via selection for hygromycin resistance.

### Media

Media used in this study consisted of 1% w/v yeast extract, 2% w/v peptone, and 2% w/v glucose (YPD). Synthetic complete (SC) media consisted of 6.7 g/L Yeast Nitrogen Base with ammonium sulfate and without amino acids, 1.77 g/L CSM-Ura (Formedium), 50 mg/L uracil (Sigma), and either 2% w/v glucose (SC-Glu), 2% w/v raffinose (SC-Raf) or 2% w/v galactose + 2% w/v raffinose w/v (SC-Gal). YPD and SC media containing Hygromycin B (Invitrogen) (200 mg/L), or Nourseothricin (clonNAT) (100 mg/L) were used for the selection of yeast transformants and plasmid maintenance. Plates of these media were made with 2% agar. Canavanine plates consisted of 6.7 g/L Yeast Nitrogen Base with ammonium sulfate and without amino acids, 0.74 g/L CSM-Arg (Formedium), 60 mg/liter L-canavanine (Sigma-Aldrich), 2% w/v agar, and 2% w/v glucose.

### CoMuTER induction

Strains were grown in 500 μL SC-Glu (plus hygromycin) for 16 h at 30 °C. Next, strains were inoculated to an $OD_{600}$ of 0.05 in SC-Raf (plus hygromycin) and grown for 12 h at 30 °C. To induce the expression of the system, strains were inoculated to an $OD_{600}$ of 0.05 in SC-Gal (plus hygromycin) and grown for 6 h at 30 °C. Non-induced control strains were grown in parallel in SC-Raf (plus hygromycin). To stop the induction and prevent further growth, cells were resuspended in SC without a carbon source. $OD_{600}$ was measured using the Infinite M200 Pro plate reader (Tecan For Life Science).

### Count of canavanine-resistant mutants after CoMuTER induction

After induction of the CoMuTER system, the $OD_{600}$ of the resuspended cultures was measured to determine cell densities (cells/mL) using the Infinite M200 Pro plate reader (Tecan For Life Science) per replicate. Between $5 \times 10^5$ and $1 \times 10^6$ cells were plated on canavanine plates per replicate. To determine the exact number of plated cells, cultures were diluted by a factor of $10^4$ and plated on YPD. Colonies were counted after 72 h of incubation at 30 °C. The number of resistant colonies was divided by the total number of cells plated, as determined from colony growth on YPD, and extrapolated to a theoretical plating of $1 \times 10^6$ cells.

### Sequencing of 12 kb region of canavanine-resistant mutants

**Amplicon generation.** For each strain and target site, 30 replicates were selected on canavanine plates (=510 samples). For each replicate two overlapping ~6 kb amplicons were generated, covering a region of ~12 kb downstream of the respective target site. PCR amplification was done using Q5 Hot-Start High-Fidelity DNA Polymerase (NEB). Primers used for amplification are listed in Supplementary Data 1, Primers. For replicates expressing untargeted CoMuTER systems, the same region as for target site 1 was amplified. The presence and quality of each amplicon were examined via absorbance measurements on NanoDrop 8000 and visually, via gel electrophoresis. The two ~6 kb amplicons per sample were pooled and sent for Nanopore sequencing at the

Neuromics Support Facility, VIB-UAntwerp Center for Molecular Neurology.

## Nanopore sequencing

PCR products were purified using AMPure XP reagent (Beckman Coulter), ratio 0.8×. Purification was performed on an automated platform (Beckman Coulter Biomek FxP). Removal of smaller molecules (e.g. primer dimers) was checked using the Agilent Fragment Analyzer 5300 and the DNF-492 Standard Sensitivity Large Fragment analysis kit. Next, each sample/amplicon was barcoded using the PCR Barcoding Expansion 1-96 kit (EXP-PBC096, Oxford Nanopore Technologies (ONT)) and the LongAmp Taq 2X Master Mix (M0287L, New England Biolabs). Barcoded samples/amplicons were purified using AMPure XP reagent (Beckman Coulter), ratio 0.8×. Purification was performed on an automated platform (Beckman Coulter Biomek FxP). Amplicons were analyzed using the Agilent Fragment Analyzer 5300 and the DNF-492 Standard Sensitivity Large Fragment analysis kit. Amplicon concentrations of each sample were quantified via Qubit 1x dsDNA High Sensitivity (Thermo Fisher Scientific). Samples were pooled in equal amounts and the resulting pool was purified using AMPure XP reagent and eluted in nuclease-free water. The purified pool was again analyzed using the Agilent Fragment Analyzer 5300 and the DNF-492 Standard Sensitivity Large Fragment analysis kit. The pooled samples were subjected to ONT library preparation using the SQK-LSK110 Ligation Sequencing kit (ONT) (input 100 fmol/pool). End repair was carried out with NEBNext Ultra II End Repair/dA-Tailing Module (E7546L, New England Biolabs). The pool was purified using AMPure XP reagent (Beckman Coulter), ratio 1×, and eluted in nuclease-free water. The concentration was quantified via Qubit 1× dsDNA High Sensitivity (Thermo Fisher Scientific). Next, adapters were added to end-repaired DNA using NEBNext Quick Ligation Module (E6056L, New England Biolabs). NEBNext Quick Ligation Reaction Buffer was replaced by ONT proprietary Ligation Buffer. The pool was purified using AMPure XP reagent (Beckman Coulter), ratio 1×, and eluted in Buffer EB (ONT). The concentration was quantified via Qubit 1× dsDNA High Sensitivity (Thermo Fisher Scientific). Finally, 10 fmol of the pool were loaded on an R9.4.1 Flongle Flow Cell (FLO-FLG001, Oxford Nanopore Technologies) with >50 available pores after Flow Cell QC.

## Nanopore sequencing data analysis

The basecalling of the Nanopore data was performed using the Guppy basecaller version v4.2.2. Analysis was performed using a pipeline integrated into genomecomb[81]. Reads were aligned to the sacCer3 genome reference (ftp://hgdownload.cse.ucsc.edu/goldenPath/sacCer3/chromosomes) using minimap2[82] and the resulting sam file was sorted and converted to a bam file using samtools[83]. Structural variants were called using sniffles[84] and npinv[85]. SNV calls and haplotype separation of the bam were performed using longshot[86]. The resulting variant sets of different individuals were combined and annotated using genomecomb[81]. Sequencing data have been deposited in the NCBI Sequence Read Archive under accession code PRJNA974923.

Variant calls with a total coverage >70 and genotype quality >50 detected on Chr 5 were selected for further analysis. Remaining variants were verified using intensive manual curation and can be found in the Source Data file, Fig. 3b, c. Only replicates that contained at least one mutation in the CAN1 gene were analyzed. The number of analyzed replicates per strain and target site is listed in Supplementary Table 8.

## Whole genome sequencing for on- and off-target analysis

**Sample preparation and sequencing.** DNA was isolated from five replicates per strain and target site using a standard zymolyase-based protocol. DNA concentrations were measured with a Qubit 2.0 and

DNA quality was checked using a NanoDrop 8000 and by gel electrophoresis. Samples were sent for paired-end DNBseq platform sequencing (BGI), with an average read length of 150 bp and an average insert size of 350 bp. Each of the samples had a minimum coverage of 80×. Sequencing data have been deposited in the NCBI Sequence Read Archive under accession code PRJNA974923.

## Whole genome sequencing data analysis

The quality of sequencing data was assessed using FastQC (version 0.11.2) and filtered using BBDuk (ktrim = r $k$ = 23 mink = 11 hdist = 1; BBMap version 38.20). Filtered reads were mapped to the reference S288c genome (assembly R64) and inserted CoMuTER sequences using bwa-mem version 0.7.17 with default settings. Mapped reads were sorted by coordinates using SortSam (Picard tools version 2.25.2) and duplicates were marked with MarkDuplicates (Picard tools version 2.25.2). SNVs and Indels were analyzed following the GATK best practice workflow "Germline short variant discovery (SNPs + Indels)". Variants were called per sample using the GATK HaplotypeCaller (version 4.1.2.0) with sample-ploidy set to 1. The resulting GVCF files of each analyzed sample were combined using GATK CombineGVCFs and GenotypeGVCFs were used to perform joint genotyping. The called SNVs and Indels were filtered separately based on INFO and/or FORMAT annotations using GATK VariantFiltration (SNVs: QUAL < 40.0; MQ < 50.0; MQRankSum < −12.5; QD < 2.0; SOR > 3.0; FS > 60.0, Indels: QD < 2.0; QUAL < 40.0; FS > 200.0; ReadPosRankSum < −20.0). Variants called in repetitive regions were filtered out as well as variants present in the ancestral strain. All remaining SNVs and Indels were verified using intensive manual curation and can be found in Supplementary Data 1, WGS SNV and WGS INDEL.

## Definition of Cas3 –APOBEC1 activity window

The activity window is defined as the area that contains 99.3% of cytidine deaminations identified on Chr 5 as defined by a boxplot (target 1: median = 31,845, lower quartile = 24,762.5, upper quartile = 34,791, end lower whisker = 11,970, end upper whisker = 49,003; target 2: median = 33,853.5, lower quartile = 28,980.5, upper quartile = 46,930.5, end lower whisker = 15,355, end upper whisker = 70,524). Cytidine deaminations outside of this range are considered outliers.

## Design and construction of lycopene biosynthesis cassette

To generate the lycopene-producing base strain, three heterologous lycopene biosynthesis genes, *geranylgeranyl diphosphate synthase* (*CrtE*, from *X. dendrorhous*), *phytoene synthase* (*CrtB*, from *P. agglomerans*) and *phytoene desaturase* (*CrtI*, from *X. dendrorhous*), as well as truncated HMG-CoA reductase (tHMG1, from *S. cerevisiae*), were inserted into CEN.PK-CB (Supplementary Table 4). CDS for *CrtE*, *CrtI*, and *tHMG1* were amplified from plasmid pLM494 (Addgene plasmid #100539[87]) and *CrtB* was ordered from BGI Genomics. Each gene was fused to native *S. cerevisiae* promoters and terminators (Supplementary Data 1, Lycopene cassette) using NEBuilder® HiFi DNA Assembly Master Mix (NEB). Each gene expression cassette (promoter, CDS, terminator) was amplified with specific primers that generate overlapping 60–90 bp spacer sequences in between cassettes using PrimeSTAR GXL DNA polymerase (TaKaRa). The same strategy was used to insert the crRNA target site directly upstream of the first expression cassette and to generate ~40 bp flanking sequences homologous to the integration site (436 bp upstream of *CAN1*). Primers sequences and crRNA target site are listed in Supplementary Data 1, Primers and Supplementary Table 7, respectively. Sequences and details on each element of the lycopene expression cassette can be found in Supplementary Data 1, Lycopene cassette. Overlapping gene cassettes were integrated into strain CEN.PK-CB (Supplementary Table 4) via CRISPR-Cas9-assisted homologous recombination using pVL382[78]. The gRNA used for CRISPR-Cas9-mediated insertion can be found in Supplementary Table 6. Yeast transformation was carried out using a DMSO-

LiAc procedure as described in Pan, X. et al.[79]. The resulting strain was used as the base strain (lycopene base strain, Supplementary Table 4) for the proof-of-concept.

## Selection of improved lycopene biosynthesis strains

The lycopene base strain was transformed with the plasmid encoding the Cas3-APOBEC1 base editor (Supplementary Data 1, Cas3-base editor plasmids) and crRNA-targeting upstream of the integrated lycopene biosynthesis pathway. After induction of the CoMuTER system (see above), cells were plated on canavanine plates (100 plates, ~1 × 10⁶ cells/plate plated) to pre-select colonies in which the system had introduced mutations. After incubation for 72 h at 30 °C and additional growth for 24 h at 22 °C colonies were visually screened for color intensity. Colonies showing more intense red coloration were selected for further analysis.

## Sequencing of selected lycopene biosynthesis strains

The lycopene cassette of the selected colonies was amplified using PrimeSTAR GXL DNA polymerase (TaKaRa) and sequenced via Sanger sequencing. Primers used for amplification and sequencing are listed in Supplementary Data 1, Primers. Detected mutations are listed in the Source Data file, Fig. 5d.

## Re-integration of mutated pathway

The mutated lycopene pathway of each selected strain was amplified using PrimeSTAR GXL DNA polymerase (TaKaRa) and re-integrated into strain CEN.PK-CB (see Supplementary Table 4) via CRISPR-Cas9-assisted homologous recombination using pV1382[78]. The gRNA used for CRISPR-Cas9-mediated insertion can be found in Supplementary Table 6. Yeast transformation was carried out using a DMSO-LiAc procedure as described in Pan, X. et al.[79]. The lycopene production of the resulting strains was analyzed via LC−MS.

## Quantification of lycopene production via LC−MS

**Lycopene extraction**. Strains were grown in 5 mL YPD for 16 h at 30 °C. Next strains were inoculated to an $OD_{600}$ of 0.05 in 50 mL YPD and grown for 72 h at 30 °C. Cultures were well-mixed and 1 mL of culture was transferred to a 2 mL light-protected Eppendorf tube (Safe-Lock Tubes, amber, Eppendorf). Samples were centrifuged (3 min at 4 °C, 3000 × g) and the supernatant was removed. Samples were washed twice with 1 mL cold (4 °C) water (centrifugation for 3 min at 4 °C, 3000 × g). After the final centrifugation, the supernatant was removed thoroughly and 400 μL of glass beads (acid-washed, 425−600 μm, Sigma Aldrich) as well as 500 μL of Acetone (≥99.8%, AnalaR NORMAPUR, VWR) were added to the sample tubes. Cells were disrupted for 60 s at maximum power using a Fastprep homogenizer (FastPrep-24, MP Biomedicals). Samples were centrifuged at 16,000 × g for 10 min at 4 °C and 200 μL of the lycopene-containing supernatant were transferred to a new 2 mL light-protected Eppendorf tube (Safe-Lock Tubes, amber, Eppendorf). The remaining cell lysate was used for four additional extraction rounds by adding 200 μL fresh Acetone (≥99.8%, AnalaR NORMA-PUR, VWR) and repeating the cell disruption and subsequent centrifugation steps. After each round, 200 μL lycopene-containing supernatant was transferred to the new 2 mL light-protected Eppendorf tube resulting in a total of 1 mL lycopene extract. The lycopene extract was stored at −20 °C for 48 h. Prior to LC−MS analysis, samples were centrifuged at 16,000 × g for 10 min at 4 °C to remove precipitates, and 300 μL of clarified supernatant was used for LC−MS analyzes at the VIB Metabolomics Core, Leuven.

## LC−MS measurements

Samples were prepared for LC−MS (liquid chromatography−mass spectrometry) measurement by transferring 100 μL of the acetone extracts to an Eppendorf tube and evaporating the acetone using a speedvac. Samples were resuspended in 100 μL acetonitrile:methanol; 70:30 (v/v). This solution was transferred to MS vials with glass inserts. The lycopene standard (1 mg/mL, stored at −20 °C) was mixed, evaporated, and resuspended in acetonitrile:methanol; 70:30 (v/v) to a concentration of 3 μM. This standard was diluted in acetonitrile:methanol; 70:30 (v/v) to prepare the other standards for a calibration curve based on 0, 0.1, 0.25, 0.5, 1, and 3 μM concentrations.

MS measurements were performed using a Vanquish LC System (Thermo Scientific) coupled to an electrospray ionization source (HESI-II Probe, Thermo Scientific) and a Q Exactive Orbitrap Focus mass spectrometer (Thermo Scientific). 10 μL sample was injected onto an Acquity UPLC-HSS T3 column (1. 8 μm; 2.1 × 150 mm, Waters) and subjected to an LC gradient method adapted from Rivera, S. et al.[88]. carried out starting with 85% solvent A (acetonitrile:methanol; 70:30,v/v) and 15% solvent B (MilliQ water) at a flow of 0.25 mL/min. These conditions were kept until 3.2 min, then followed by a linear increase to 100% solvent A at 4.8 min. These conditions were kept until 11.2 min, then followed by a linear increase of the flow to 0.3 mL/min at 12.8 min. These conditions were kept until 21.6 min, then followed by a linear decrease of the flow to 0.25 mL/min and a linear decrease to 85% solvent A at 23.9 min. The column was then equilibrated at these conditions until 29.2 min. The column temperature was kept constant at 32 °C. The mass spectrometer operated in full scan (range [700,000−10,500,000]) and positive mode using a spray voltage of 3.5 kV, capillary temperature of 320 °C, sheath gas flow rate at 45, auxiliary gas at 0, sweep gas at 2. AGC target was set at 3.0E + 006 using a resolution of 70,000. Data collection was performed using the Xcalibur software (Thermo Scientific). The most reproducible ion selected for further data treatment was the [M•]+ cationic radical ion. The data analyses were performed by integrating the peak areas (El-Maven−Polly-Elucidata). The measurement was preceded MS by 10 injections of a mock sample containing only acetonitrile:methanol; 70:30 (v/v) to equilibrate the instrument and column, and a QC obtained by mixing small aliquots of all sample solutions was run regularly in order to check for and, if required, correct for potential signal drift.

## RNA extraction and quantitative real-time PCR

The six selected lycopene production strains and the base strain were grown in 1 ml SC-Glu for 16 h at 30 °C (three biological replicates per strain). Next, strains were inoculated to an $OD_{600}$ of 0.05 in 3 ml fresh SC-Glu and grown until an $OD_{600}$ of 0.6 was reached. $OD_{600}$ was measured using the Infinite M200 Pro plate reader (Tecan For Life Science). Total RNA was extracted from 1 ml of culture using the MasterPure Yeast RNA Purification Kit (LGC Biosearch Technologies) and cDNA synthesis was carried out using QuantiTect reverse transcription kit (Qiagen) according to the instructions provided by the supplier. Obtained cDNA was used for quantitative real-time PCR (qPCR). Primers for quantitative real-time PCR (Supplementary Data 1, Primers) were designed using IDT's PrimerQuest Tool followed by BLAST analysis against the *S. cerevisiae* genome to ensure specificity.

qPCR was performed using Power SYBR™ Green PCR Master Mix according to the manufacturer's protocol (Applied Biosystems) and a StepOnePlus Real-Time PCR System (Applied Biosystem). Each reaction contained a final volume of 20 μL (10 μL SYBR Green Master Mix, 4 μL nuclease-free water, 1 μL of each primer (10 μM), 4 μL (75 ng) of cDNA). Each sample was tested in triplicate. The thermal protocol consisted of AmpliTaq Gold activation for 10 min at 95 °C followed by 40 cycles of 15 s denaturation at 95 °C, 15 s annealing at 66 °C and 1 min extension at 72 °C. After thermal cycling, a melt curve protocol was run: 15 s at 95 °C, 1 min at 60 °C, and 15 s at 95 °C (ramp rate 0.3 °C/s). As reference genes *TAF10* and *ALG9* were used[89]. CT values for the examined genes (*tHMG1, crtE, crtI, crtB*) and reference genes were calculated by the StepOne Software (Supplementary Data 1, Raw CT values). We calculated the average CT value for the two reference

genes and used this value to calculate the dCT values for the examined genes by subtracting it from the CT value of the examined genes. The mean $2^{-\Delta\Delta CT}$ values (fold change) were calculated using the $2^{-\Delta\Delta CT}$ method[90] and used as the indicator of the expression of the examined genes in lycopene production strains relative to the base strain. Calculated dCT, ddCT, and $2^{-\Delta\Delta CT}$ values are provided in the Source Data file and Supplementary Fig. 12.

## Targeting of *SEC14* locus

To test the activity of CoMuTER at a different genomic location, we targeted the system to a site 386 bp downstream of the *SEC14* CDS (Supplementary Table 7). The essential gene *SEC14* encodes a phosphatidylinositol transfer protein that represents the sole target of a class of small molecule inhibitors termed nitrophenyl(4-(2-methoxyphenyl)piperazin-1-yl)methanones (NPPMs) used against pathogenic fungi. As mutations in *SEC14* can confer resistance to these inhibitors, CoMuTER was employed to generate resistant phenotypes. We used the NPPM 6748-481 purchased from Axon Medchem (SMI 481, CAS 432020-20-7).

To determine a suitable inhibitor concentration, the growth of the parental strain CEN.PK-CB was examined in the presence of increasing concentrations of NPPM 481. CEN.PK-CB, as well as a positive control containing a resistance-conferring mutation in *SEC14* (Y111A), were pre-grown in SC-Glu and adjusted to a density of 250,000 cells/ml. Then, 4 μl (=1000 cells) of this culture were plated in 5-fold serial dilution on SC-Glu supplemented with 0, 0.5, 1, 2, 3, 4, 8, 12, 16, 20, and 30 μM NPPM 481 dissolved in DMSO. CEN.PK-CB was able to grow on a medium containing up to 1 μM NPPM 481, while the positive control was able to grow on a medium containing >20 μM NPPM 481. NPPM resistance is strongly dependent on the employed background strain, as strain S288c-CB was able to grow on a medium containing up to 8 μM NPPM 481 while the respective positive control grew on all tested concentrations. Based on these results, we chose an NPPM 481 concentration of 3 μM for further experiments.

CEN.PK-CB was transformed with a plasmid encoding the Cas3-APOBEC1 base editor and the crRNA targeting 386 bp downstream of the *SEC14* stop codon. As a control, CEN.PK-CB was transformed with a plasmid containing the Cas3-APOBEC1 base editor without crRNA (untargeted Cas-APOBEC1). The two generated strains, as well as the parental CEN.PK-CB, was grown in three biological replicates. After induction of the CoMuTER system (see above), each biological replicate was plated in three technical replicates on SC-Glu plates supplemented with 3 μM NPPM 481 (~50,000 cells/plate, 9 plates per strain). Colony formation was assessed after 48 h of incubation at 30 °C. For the strain expressing the targeted Cas3-APOBEC1 base editor, 2 colonies per plate (18 colonies in total) were selected for subsequent Sanger sequencing of the *SEC14* locus (171 bp upstream to 597 bp downstream of the start codon) to identify introduced mutations. Sequencing data have been deposited in the NCBI Sequence Read Archive under accession code PRJNA974923.

## Statistics and reproducibility

To analyze statistical differences in the number of canavanine-resistant colonies identified per strain and target site the data were transformed to proportions of cells containing a knock-out mutation and analyzed using a generalized linear mixed-effects model with a binomial link function and biological replicates as random effects (R package lme4[91], function glmer). To identify strains with significant differences from the control strain a two-sided post-hoc test with Sidak adjustment was used (R package lsmeans[92], function lsmeans).

To analyze statistical differences in the number of cytidine deamination identified per strain and target site the data were first fitted with a generalized linear mixed-effects model with a Poisson link function (Gaussian link function for "on-target fold-increase", Fig. 4b) and biological replicates as random effects (R package lme4[91], function

glmer). To identify strains with significant differences to the control strain a two-sided post-hoc test with Sidak adjustment was used (R package lsmeans[92], function lsmeans).

Statistical significance of differences between selected lycopene-producing strains and the lycopene base strain was assessed using an unpaired two-sided two-sample *t*-test (R package stats, function *t*.test).

To analyze statistical differences in the expression of the four introduced lycopene production genes between the selected strains and the lycopene base strain the data were fitted with a linear model (R package stats, function lm). To identify strains with significant differences to the base strain emmeans (package emmeans[93], function emmeans) with dunnettx adjustment (a close approximation to the Dunnett adjustment) was used.

To verify the reproducibility of experiments, experiments were performed using biological replicates. Detailed information on the respective number of experimental replicates can be found in the individual figure legends. In Fig. 5c an exemplary plate showing variation in colony color and size after one round of CoMuTER mutagenesis is shown. In total 100 plates were screened to identify 6 colonies with increased red coloration compared to the parental strain.

## Reporting summary

Further information on research design is available in the Nature Portfolio Reporting Summary linked to this article.

## Data availability

Data supporting the findings of this work are available within the paper and its Supplementary Information files. The whole genome sequencing data and the Oxford Nanopore sequencing data generated in this study, as well as the Sanger sequencing data of the SEC14 locus, have been deposited in the NCBI Sequence Read Archive under accession code PRJNA974923. All yeast strains and plasmids described in this work are available upon request. Source data are provided with this paper.

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

## Acknowledgements

Special thanks to Dr. Charles Margarit and Dr. Karin Voordeckers for their valuable feedback on the manuscript, Lloyd Cool for his continual advice on the statistical analyses, Dr. Brigida Gallone and Dr. Polina Novikova for their support with the WGS data analysis, Dr. Peter Bircham for the support with image processing, and all Verstrepen lab members for valuable discussions. Thanks to the Neuromics Support Facility, VIB-UAntwerp Center for Molecular Neurology, and the VIB Metabolomics Core, Leuven for help with experimental assays. Parts of Figs. 1a and 2a were created with BioRender.com. A.Z. was supported by a Vlaams Instituut voor Biotechnologie (VIB) Ph.D. fellowship. J.E.P.-V. and C.C. are supported by Ph.D. fellowships from FWO (1S25923N, 1SC2422N| 1SC2420N). J.S. acknowledges financial support from FWO through a postdoctoral fellowship (12W3921N|12W3918N). The research was supported by an FWO project (G019223N) and an iBOF project (IBOF/21/092). Research in the lab of K.J.V. is supported by KU Leuven C1 Financing, VIB, VLAIO, FWO, and iBOF.

## Author contributions

A.Z., A.G., J.S., Y.Vd.P., and K.J.V. conceptualized the study. A.Z., J.E.P.-V., and C.C. designed the experiments. A.Z. and J.E.P.-V. performed the experiments. A.Z., J.E.P.-V., C.C., A.G., J.S., Y.v.d.P., and K.J.V. contributed to the data analysis. A.Z., J.E.P.-V., C.C., A.G., J.S., Y.v.d.P., and K.J.V. wrote and reviewed the manuscript.

## Competing interests

The authors declare no competing interests.
