## [Peer Review File · Nature Communications]

Reviewers' Comments:

Reviewer #1:

Remarks to the Author:

Reviewer's comments:

In this manuscript, Zimmermann et al. describe CoMuTER (confined mutagenesis using a type I-E CRISPR-Cas system), a novel tool for random mutagenesis. This tool combines Cas3 from the class 1 type I-E CRISPR-Cas system of *E. coli* and cytidine deaminase. CoMuTER is the first example of a class 1 type I-E CRISPR-Cas system being used for random mutagenesis; however, its importance needs to be further demonstrated by applying it to human cells, animal cells, or models other than *E. coli*. In addition, in a situation wherein research using the base editing system in *E. coli* is being actively undertaken, the importance of broadening the editing range by random mutagenesis rather than high-target specificity does not seem to be an acceptable option. It is also necessary to plan how the reduction of off-target variations can be achieved. There are several major concerns regarding this manuscript, which are summarized below.

1. To highlight the importance of this novel tool, it is vital to apply random mutagenesis in mammalian cell lines or animals.
2. Have you checked whether there is any difference in the base editing efficiency based on the arrangement position of Cas3 or deaminase?
3. Additional experiments are required on how to remove or control off-target mutations that occur in unwanted locations by this system.
4. You need to revise 'class 1 type I CRISPR-Cas' to 'class 1 type I-E CRISPR-Cas' (line83).

Reviewer #2:

Remarks to the Author:

Zimmermann et al provide a manuscript detailing the interesting use of the Cas3 enzyme inducible and targetable, in vivo mutagenesis of genomic loci in *S. cerevisiae* yeast. Controllable targeted mutagenesis in vivo is a growing area of focus for researchers in microbiology, synthetic biology and biotechnology, especially when coupled with directed evolution. The Orthorep system of Chang Liu's team is the current state-of-the-art in yeast with very high mutation rates of genes placed on special plasmids inside the yeast. However, it is cumbersome to work with and doesn't allow you to simply target a section of the genome. Other methods have also been developed that are described in the introduction of this paper.

The approach here sees Cas3 from *E. coli* used in yeast to target genomic loci of interest by co-expression of crRNA sequences with spacers designed to target the loci sequence of interest, similar to how gRNAs target DNA for cleavage when using Cas9. The crucial difference here is that Cas3 is fused to a cytidine deaminase so that as it goes about its usual job processing the targeted DNA it also does semi-random base editing which leads to these loci getting up to 350-fold more mutations than the rest of the genome. The authors determine this by CAN1 mutation assays, also showing the long lengths of genome that can be mutated (kb-scale) with a higher rate using this approach. The off-target effects are also investigated before they go on to demonstrate an industrial biotechnology use case where a heterologous lycopene biosynthesis pathway construct is mutated to reveal mutations that alter its performance.

Overall this is an interesting and novel addition to the growing literature in this area, providing a new molecular tool that looks simple to use and will give enhanced mutation at targeted genomic loci. It is well-written and well presented and leaves me with only minor comments:

Comments

- Line 107-110

The authors mentioning the necessity of integrating five cascade enzymes (Cas5, Cas6, Cas7, Cas8 and Cas11) into the yeast genome to generate the background strain. Can the authors please further elaborate on the design and the stoichiometry considerations as these are just briefly touched on in the main text and will be important for anyone wanting to extend from this

work.

- Figure 2

Please provide the actual average counts for each bar. Especially the numbers for uninduced conditions, which are hard to estimate for the figure. This would also help to better understand the differences in counts between induced and uninduced conditions

- Figure 2

While most constructs lead to negligible colony counts under uninduced conditions, a high amount of colonies is obtained in the case of Cas3-CDA1. Can the authors provide an explanation/theory for this effect?

- Figure 2

Can the authors elaborate on which sequences they used as 'untargeted' sequences for the system? And how do the authors explain rather high CAN1 knock-out rates using these sequences in comparison to the best performing target sequences (approx. 10 % for Cas3-Apobec and even 30 % in case of Cas3-CDA1)?

- Line 335-342

This section gives the impression that the successful use of the CoMuTER system is somewhat dependent on the application of a co-selection system. As also stated in lines 338/339 using CAN1 'to enable parallel selection of cells with active CoMuTER'. The dependency on a certain integration locus/co-selection system however significantly reduces the overall applicability of the system. Can the authors please elaborate on the single cell level activity of the CoMuTER system, since the results give the impression that the system is active in just a fraction of the whole population. Further experiments using a different genomic locus without parallel CAN1 knock out selection would therefore be very helpful to judge the overall efficiency of the system, which could be very low without selection.

- Line 339

The term 'basal activity' is not really a good phrase to use since the introduction of the pathway should lead to the production of lycopene. This rather sounds like unintended production.

- Figure 5

Was qRT PCR analysis of the expression profiles pathway genes performed? It would be extremely helpful to have data on the relative expression level of each gene within the 7 strains (base strain + 6 improved strains) to deconvolute improved enzyme variants from gene level alterations (mutations in the promoter region). This is illustrated by comparing strain 5 and 6, which share the crtB S228F variant, but in case of strain 6 an additional mutation within the promoter region further elevates lycopene levels. A few small further experiments would really shed light on this issue.

- **Testing the activity of the CoMuTER system at a different genomic locus**

Reviewer 2 raised the question whether the system is dependent on a specific target-location or co-selection marker, in this case *CAN1*. We have now added an experiment in which the CoMuTER system is targeted to the essential gene *SEC14*, located on chromosome 13. We used CoMuTER to introduce mutations that confer resistance to a small molecule inhibitor (NPPM 481) that blocks the activity of the wild-type Sec14 phosphatidylinositol transfer protein. We sequenced a selection of the resistant variants and found various cytidine deaminations throughout the *SEC14* locus that enabled survival (Supplementary Table 3). These data demonstrate the capabilities of CoMuTER in a different genomic context without the need for *CAN1* co-selection.

- **Assessing the relative expression levels of the lycopene pathway genes in the improved strains**

Reviewer 2 asked about the relative expression of the lycopene pathway genes within the improved strains and the base strain to shed light on potential expression level alterations caused by mutations in promoter and terminator regions of the improved strains. We have now assessed the relative expression level of each gene (*tHMG1*, *crtE*, *crtI* and *crtB*) within the 6 selected strains and the base strain via qPCR (Supplementary Fig. 5). We did not find a significant difference in relative gene expression between the improved strains and the base strain. This suggests that increases in lycopene production are either caused by alterations on the translational level or mutations in the coding regions of *tHMG1*, *crtE*, *crtI* and *crtB*, causing improved enzyme variants.

Below, we address all comments of the reviewers in detail (blue font, in-between the reviewer's comments).

Best wishes,
Kevin

REVIEWER COMMENTS

Reviewer #1 (Remarks to the Author):

In this manuscript, Zimmermann et al. describe CoMuTER (confined mutagenesis using a type I-E CRISPR-Cas system), a novel tool for random mutagenesis. This tool combines Cas3 from the class 1 type I-E CRISPR-Cas system of *E. coli* and cytidine deaminase. CoMuTER is the first example of a class 1 type I-E CRISPR-Cas system being used for random mutagenesis; however, its importance needs to be further demonstrated by applying it to human cells, animal cells, or models other than *E. coli*. In addition, in a situation wherein research using the base editing system in *E. coli* is being actively undertaken, the importance of broadening the editing range by random mutagenesis rather than high-target specificity does not seem to be an acceptable option. It is also necessary to plan how the reduction of off-target variations can be achieved. There are several major concerns regarding this manuscript, which are summarized below.

We thank the reviewer for her/his assessment of our work. We do want to point out that although we did use a type I-E CRISPR-Cas system from *E. coli*, we did not develop our tool in *E. coli* as the reviewer seems to think, but rather in *S. cerevisiae*, a eukaryotic model microbe that has traditionally been used as a cell factory in various applications, and for which the use is predicted to further increase in the future. As such, we have already shown that our system works in eukaryotic cells.

Comments

1. To highlight the importance of this novel tool, it is vital to apply random mutagenesis in mammalian cell lines or animals.

Thank you for this suggestion. While applying CoMuTER in mammalian cell lines could be considered for a follow-up study, and while we are quite certain that the tool would work, it goes beyond the scope of this work. The CoMuTER system was developed as a tool to improve cell factories based on *S. cerevisiae*, an important eukaryotic microbial chassis in biotechnology. Moreover, we have tested our system in two different *S. cerevisiae* strains (S288C and CEN.PK, the two most used chassis strains). Developing a tool for pathway optimization in microbes is important and useful because microbes are increasingly used for sustainable production of a broad range of industrially relevant compounds. In fact, many previously reported mutagenesis tools have been solely developed and tested in microbes¹⁻⁷.

2. Have you checked whether there is any difference in the base editing efficiency based on the arrangement position of Cas3 or deaminase?

The reviewer raises an interesting point that we have considered and explored ourselves to some degree already. Specifically, we have tested two separate deaminases fused to Cas3, *rAPOBEC1* and *pmCDA1*. *rAPOBEC1* was N-terminally fused to Cas3 while *pmCDA1* was fused to the C-terminus of Cas3. We know that both deaminases are active in *S. cerevisiae*, as both Cas3-CDA1 and Cas3-APOBEC1 caused a significantly increase in genome-wide cytidine deaminations (Fig. 4a) and knock-out mutations in *CAN1* (Fig. 2c). Yet, in contrast to N-terminally fused *rAPOBEC1*, C-terminally fused *pmCDA1* did not cause a significant increase in cytidine deaminations within a larger window downstream of the target site (Fig. 3b). Therefore, we believe that the C-terminal fusion of *pmCDA1* is impairing the helicase activity of Cas3. We briefly discuss this in the manuscript (line 447-451) :

“In contrast to APOBEC1, which was N-terminally fused to Cas3, CDA1 was fused to the C-terminus of Cas3 (Fig. 1, orientation based on reported fusion with dCas9^{41,42}). This orientation might sterically hinder the helicase activity of Cas3, while still allowing CDA1 to be active in the region proximal to the targeted site. Cas3-CDA1 is recruited to the target site where CDA1 can introduce cytidine deaminations, but sterically prevents DNA unwinding.”

As both cytidine deaminases are active, but only the N-terminal fusion construct led to a significant increase in deaminations further downstream of the target site, we did not see the need to iteratively examine base editor construction designs by alternating the fusion direction of the different deaminases. Instead, we chose to focus our efforts on demonstrating the applicability of the designed system for pathway optimization by employing it to enhance lycopene production.

3. Additional experiments are required on how to remove or control off-target mutations that occur in unwanted locations by this system.

While we appreciate the reviewer’s feedback, we do not think that reduction of off-target mutations is required for the intended application of the system. First, it is important to note that background mutation levels are never zero (not even in wild-type cells). Although we found an increase in the number of off-target mutations upon expression of the CoMuTER system compared to wild-type cells, the on-target activity is ~350-fold greater. Thus, CoMuTER significantly increases the probability of introducing mutations in the targeted region.

Moreover, promiscuous cytidine deaminase activity leading to increased off-target mutagenesis is also observed for more traditional Cas9-based base editors⁸⁻¹¹. Similarly, previously developed tools for *in-vivo* random mutagenesis that employ *rAPOBEC1* or *pmCDA1* also report an increase in the frequency of mutations at an off-target locus¹². We have now elaborated on this in the revised manuscript (line 473-478):

“[...] strains expressing a cytidine deaminase fused to Cas3 or dnCas3 showed a significant increase in the number of off-target cytidine deaminations compared to the parental S288c strain (Fig. 4a). This increase is independent of the crRNA-guided targeting of the Cas3-base editor. Guide RNA-independent off-target mutagenesis has already been reported for cytosine base editors and is likely caused by the intrinsic DNA affinity of the deaminase domain^{71,72}.”

Also, as variants with improved phenotypes must be (screened and) selected after CoMuTER activity, those strains harboring detrimental off-target mutations will be naturally sorted out. Together with the other benefits of CoMuTER compared to existing tools for random mutagenesis, we are convinced that this relatively low number of off-target mutations does not pose a problem for its applicability.

4. You need to revise 'class 1 type I CRISPR-Cas' to 'class 1 type I-E CRISPR-Cas' (line83).

We agree with the reviewer and have revised the wording accordingly.

Reviewer #2 (Remarks to the Author):

Zimmermann et al provide a manuscript detailing the interesting use of the Cas3 enzyme inducible and targetable, in vivo mutagenesis of genomic loci in *S. cerevisiae* yeast. Controllable targeted mutagenesis in vivo is a growing area of focus for researchers in microbiology, synthetic biology and biotechnology, especially when coupled with directed evolution. The Orthorep system of Chang Liu's team is the current state-of-the-art in yeast with very high mutation rates of genes placed on special plasmids inside the yeast. However, it is cumbersome to work with and doesn't allow you to simply target a section of the genome. Other methods have also been developed that are described in the introduction of this paper.

The approach here sees Cas3 from *E. coli* used in yeast to target genomic loci of interest by co-expression of crRNA sequences with spacers designed to target the loci sequence of interest, similar to how gRNAs target DNA for cleavage when using Cas9. The crucial difference here is that Cas3 is fused to a cytidine deaminase so that as it goes about its usual job processing the targeted DNA is also does semi-random base editing which leads to these loci getting up to 350-fold more mutations than the rest of the genome. The authors determine this by CAN1 mutation assays, also showing the long lengths of genome that can be mutated (kb-scale) with a higher rate using this approach. The off-target effects are also investigated before they go on to demonstrate an industrial biotechnology use case where a heterologous lycopene biosynthesis pathway construct is mutated to reveal mutations that alter its performance.

Overall this is an interesting and novel addition to the growing literature in this area, providing a new molecular tool that looks simple to use and will give enhanced mutation at targeted genomic loci. It is well-written and well presented and leaves me with only minor comments:

We would like to thank the reviewer for his/her kind words. We would also like to thank the reviewer for the constructive criticism and for indicating where the manuscript lacked information or clarity. Below and in the revised manuscript, we have addressed all the reviewer's comments. We have also added new experimental data on (i) the activity of the CoMuTER system at a different genomic locus without *CAN1* selection and (ii) the relative expression levels of the lycopene pathway genes.

Comments

- Line 107-110

The authors mentioning the necessity of integrating five cascade enzymes (Cas5, Cas6, Cas7, Cas8 and Cas11) into the yeast genome to generate the background strain. Can the authors please further elaborate on the design and the stoichiometry considerations as these are just briefly touched on in the main text and will be important for anyone wanting to extend from this work.

We thank the reviewer for pointing this out. As suggested, we have now added further information about our design and stoichiometry considerations in the methods section "Design and construction of the CoMuTER system" of the revised manuscript (line 552-565):

"The class 1 type I-E CRISPR-Cas Cascade complex consists of one copy of Cas5, Cas6 and Cas8, two copies of Cas11, and six copies of Cas7. [Genes encoding Cascade subunits (*Cas5*, *Cas5*, *Cas7*, *Cas8*, *Cas11*) were taken from *E. coli* K-12 MG1655 and sequenced optimized for expression in *S. cerevisiae* (IDT Codon Optimization Tool).] For correct subcellular localization, two copies of the SV40 nuclear localization sequences (NLS) were introduced N- or C-terminally of each gene (depending on the availability of free N- or C-termini of each subunit in the Cascade complex). [Gene expression was controlled by native *S.*

cerevisiae promoters and terminators.] For Cas7, Cas8 and Cas11 constitutive promoters with different strengths were chosen, while expression of Cas5 and Cas6 was driven by the inducible *GAL1-10* promoter. Inducibility of Cas6 and Cas5 (responsible for crRNA processing and binding of the 5' handle of the mature crRNA, respectively) prevents the formation of the Cascade complex under normal growth conditions. We employed the *GAL1-10* promoter for only two Cascade subunits to reduce competition for transcription factors and to lower the risk of recombination between similar promoter sequences. To account for the higher copy number of Cas7 in the assembled Cascade complex (six copies) we selected the strong *TDH3* promoter and *PRM9* terminator.”

We also provide the sequences of each element of the designed gene constructs in “Supplementary Data 1, Cascade subunits” for a precise overview of each Cascade subunit, including flanking regions, promoter and terminator sequences and nuclear localization sequences. The respective integration site of each gene construct as well as the Cas9 guide RNA used for integration are specified in Supplementary Table 5 and 6 respectively.

- Figure 2

Please provide the actual average counts for each bar. Especially the numbers for uninduced conditions, which are hard to estimate for the figure. This would also help to better understand the differences in counts between induced and uninduced conditions

Thank you for pointing this out. We have added a table, Supplementary Table 1, that provides the exact numbers for each bar and refer to it in the caption of Figure 2.

- Figure 2

While most constructs lead to negligible colony counts under uninduced conditions, a high amount of colonies is obtained in the case of Cas3-CDA1. Can the authors provide an explanation/theory for this effect?

We agree with the reviewer that this increase in colony number for Cas3-CDA1 under uninduced conditions stands out. We hypothesize that the observed activity is caused by a combination of (i) leaky expression from the *GAL1-10* promoter driving the expression of Cas3-CDA1 and (ii) the highly active CDA1 cytidine deaminase. For (i), the *GAL1-10* promoter, while tightly regulated, has been reported to be inherently leaky^{13,14}. Moreover, cells were pre-grown in raffinose to alleviate glucose-repression and enable swift galactose-induction. This lack of glucose-repression could enhance weak expression from the *GAL1-10* promoter. As for (ii), the high number of off-target mutations for fusion constructs featuring *pmCDA1* suggest that *pmCDA1* is highly active. Comparatively high activity of *pmCDA1* relative to other cytidine deaminases (i.e. *rAPOBEC1*) has also been reported in other studies^{12,15}. This high activity in combination with leaky expression from the *GAL1-10* promoter might cause the increase in canavanine-resistant colonies under uninduced conditions.

We have now added this hypothesis to our revised manuscript (line 196-198):

“The slightly increased number of colonies in strains expressing Cas3-CDA1 under uninduced conditions is possibly caused by a combination of leaky expression from the *GAL1-10* promoter^{49,50} driving the expression of Cas3-CDA1, and the highly active *pmCDA1* cytidine deaminase^{20,42}.”

- Figure 2

Can the authors elaborate on which sequences they used as 'untargeted' sequences for the system? And how do the authors explain rather high CAN1 knock-out rates using these sequences in comparison to the best performing target sequences (approx. 10 % for Cas3-Apobec and even 30 % in case of Cas3-CDA1)?

We thank the reviewer for these questions. The untargeted conditions feature a plasmid without integrated 32 bp spacer sequence in the crRNA expression cassette. Instead, this cassette contains the 20 bp "placeholder" DNA which contains the BsmBI restriction sites and is not present in the *S. cerevisiae* genome. We now added this information to the results section of the revised manuscript (line 190-191) and the caption of Figure 2 (line 215-217).

Like reviewer 2, we have also wondered why the untargeted Cas3 base editor resulted in relatively high numbers of colonies, but the exact mechanism remains unclear. As the number of off-target mutations identified via whole-genome sequencing is substantially lower (~350-fold) than the number of on-target mutations, we do not believe that the off-target activity of the respective deaminase is causing the observed increase in colonies. Plating-based experiments typically show some degree of variability which might partially explain the slightly higher number of colonies observed when the base editors are not targeted. Another potential reason can be the occurrence of a *CAN1*-knock out mutation at an early time point during growth which propagates to a large proportion of the population (i.e. the principle of fluctuation assays as first reported in the Luria-Delbrück studies¹⁶).

- Line 335-342

This section gives the impression that the successful use of the CoMuTER system is somewhat dependent on the application of a co-selection system. As also stated in lines 338/339 using *CAN1* 'to enable parallel selection of cells with active CoMuTER'. The dependency on a certain integration locus/co-selection system however significantly reduces the overall applicability of the system. Can the authors please elaborate on the single cell level activity of the CoMuTER system, since the results give the impression that the system is active in just a fraction of the whole population. Further experiments using a different genomic locus without parallel *CAN1* knock out selection would therefore be very helpful to judge the overall efficiency of the system, which could be very low without selection.

We thank the reviewer for their insightful comments. We would like to clarify that the system is not dependent on a specific target-location or co-selection marker. In the original experiments included in our paper, the CoMuTER system was targeted to different sites around the *CAN1* gene, located on chromosome 5. We have now included an experiment in which the CoMuTER system is targeted to the essential gene *SEC14*, located on chromosome 13 (Chr. XIII, 424989-426059). Importantly, we find that CoMuTER was able to introduce cytidine deaminations in *SEC14* that confer resistance to a small molecule inhibitor (NPPM 481) that normally blocks the activity of the Sec14 phosphatidylinositol transfer protein¹⁷. These data demonstrate the capabilities of CoMuTER in a different genomic context. The section in our revised manuscript (lines 402-427) now reads:

"To test the activity of CoMuTER at a different genomic location, we targeted a site 386 bp downstream of the essential gene *SEC14*. *SEC14* encodes a phosphatidylinositol transfer protein which is essential for intracellular lipid metabolism⁶⁰. Importantly, Sec14 is the only target of a class of small molecule inhibitors termed nitrophenyl(4-(2-methoxyphenyl)piperazin-1-yl)methanones (NPPMs), used to inhibit growth of pathogenic fungi. Several mutations in *SEC14* have been reported to confer resistance to the NPPM 481,

while maintaining Sec14 function^{12,61,62}, making it an interesting target to test the ability of the CoMuTER system to generate resistant variants.

After induction of the CoMuTER system, cells were plated on medium containing 3 μ M NPPM 481 to select for variants containing resistance-conferring mutations (see Methods for further information about the chosen concentration and experimental setup). We found an average of 10 colonies per plate (corresponding to 0.02% of plated cells) that were able to grow in the presence of the small-molecule inhibitor (Supplementary Table 2). Control strains expressing either no Cas3-base editor (CEN.PK-CB) or an untargeted Cas3-base editor (Cas3-APOBEC1, untargeted) showed an average of \sim 0.22 and \sim 1.66 colonies per plate respectively (corresponding to 0.00044% and 0.00332% of plated cells respectively). We subsequently sequenced the *SEC14* locus (171 bp upstream to 597 bp downstream of the start codon) of 18 colonies that showed resistance to NPPM 481 after CoMuTER activity. Individual colonies contained between 1 and 7 cytidine deaminations with an average of \sim 2 cytidine deaminations per colony (Supplementary Table 3). Importantly, all identified mutations (37 in total) were cytidine deaminations. Among these, 18 caused amino acid substitutions, resulting in seven unique amino acid changes: H112Y, E150K, V154I, S173L, S183F, G210S and S222F. The remaining 19 mutations caused either synonymous changes or were outside the CDS (Supplementary Table 3). Of the seven unique amino acid substitutions, residue H112, V154, S173 and G210 have been previously reported to confer resistance to NPPM 481^{12,62,63} further strengthening our screening results. These data demonstrate the capabilities of CoMuTER in a different genomic context and its ability to introduce resistance-conferring mutations in the essential gene *SEC14*.”

As demonstrated in this experiment, phenotypes that can be linked to a growth advantage do not require co-selection. Similarly, co-selection is not needed if the examined phenotype can be screened via high-throughput methods, like for example fluorescence-activated cell sorting (FACS).

Unbiased, single cell level activity of the CoMuTER system is challenging to determine, as it would require sequencing of thousands of colonies. Based on the data shown in Figure 2, the number of canavanine-resistant colonies, which serve as a proxy for system activity (efficiency), is \sim 0.35%. The actual number of cells with an active system is likely higher, as selection on canavanine-containing medium is biased by the requirement of knock-out mutations in *CAN1*. Nevertheless, based on the average number of cells in common volumes of yeast cultures, a low per-cell activity does not prevent the successful use of the system in microbial cells – this is clearly demonstrated by our proof of concept experiments.

- Line 339

The term ‘basal activity’ is not really a good phrase to use since the introduction of the pathway should lead to the production of lycopene. This rather sounds like unintended production.

We agree with the reviewer and have changed the wording. The sentence now reads as follows: “The resulting lycopene base strain yielded orange-colored colonies, indicating functional expression of the introduced pathway (Fig. 5e, bottom right colony).”

- Figure 5

Was qRT PCR analysis of the expression profiles pathway genes performed? It would be extremely helpful to have data on the relative expression level of each gene within the 7 strains (base strain + 6 improved strains) to deconvolute improved enzyme variants from gene level alterations (mutations in the promoter

region). This is illustrated by comparing strain 5 and 6, which share the *crtB* S228F variant, but in case of strain 6 an additional mutation within the promoter region further elevates lycopene levels. A few small further experiments would really shed light on this issue.

We thank the reviewer for this suggestion and have now assessed the relative expression level of the introduced lycopene pathway genes within the 6 selected strains and the base strain via quantitative real-time PCR (qPCR, see Methods). None of the selected strains showed a significant difference in relative expression of the examined genes compared to the base strain (see Supplementary Fig. 5). This suggests that increases in lycopene production are either caused by alterations on a translational level or mutations in the coding regions of *tHMG1*, *crtE*, *crtI* and *crtB*, causing improved enzyme variants. Moreover, the differences in lycopene production between strain 4, 5 and 6, which share the *crtB* S228F variant, are not significant (unpaired two-samples t-test). This suggests that *crtB* S228F is the sole driver of the increased lycopene production in the three best performing strains compared to the base strain. Additional mutations in strain 4 and 6 do not significantly affect lycopene production.

These data have now been added to the results section of the revised manuscript (line 370-378):

“To determine whether the introduced mutations affect gene expression levels in the selected strains, relative expression of the lycopene pathway genes was determined via quantitative real-time PCR (qPCR). None of the selected strains showed a significant difference in relative mRNA expression of the examined genes compared to the base strain (Supplementary Fig. 5). [Strikingly, the three best performing strains, strain 4, 5 and 6, contained the same mutation in the *crtB* gene causing a serine to phenylalanine substitution at position 228.] The differences in lycopene production between these strains are not significant (unpaired two-samples t-test). Moreover, strain 5 did not contain any additional mutation. This suggests that *crtB* S228F is the sole driver of the increased lycopene production in the three best performing strains.”

The data are discussed in line 506-511 of the revised manuscript:

“Although we identified mutations in the promoter and terminator regions of the introduced lycopene pathway genes in the selected strains, relative gene expression levels were unchanged when analyzed by qPCR (Supplementary Fig. 5). This suggests that increases in lycopene production are either caused by alterations on a translational level or changes in the catalytic activity of *tHMG1*, *crtE*, *crtI* and *crtB*. Indeed, we were able to identify a mutation in *crtB*, S228F, that caused a two-fold increase in lycopene production.”

Supplementary Fig. 5 Mutated strains and base strain do not show significant differences in lycopene gene expression. To assess whether mutations introduced in the lycopene pathway of the selected strains effect gene expression, the relative expression of the introduced genes (*tHMG1*, *crtE*, *crtI*, *crtB*) was compared to the respective gene expression in the base strain. Relative gene expression was measured via qPCR using *TAF10* and *ALG9* as reference genes (see methods). **a-d** Mean fold change of *tHMG1*, *crtE*, *crtI* and *crtB* expression in selected strains 1-6 relative to the base strain. Data are presented as mean \pm 95% CI of N = 3 biological replicates. Data were fitted using a linear model with a post-hoc Dunnett test to identify strains with significant difference to the base strain ($p \leq 0.05$). P-values can be found in Supplemental Data 1, p-values. Individual data points can be found in Supplemental Data 1, mean fold change.

References

1. Wang, H. H. *et al.* Programming cells by multiplex genome engineering and accelerated evolution. *Nature* **460**, 894–898; 10.1038/nature08187 (2009).
2. Garst, A. D. *et al.* Genome-wide mapping of mutations at single-nucleotide resolution for protein, metabolic and genome engineering. *Bio/Technology* **35**, 48–55; 10.1038/nbt.3718 (2017).
3. Ravikumar, A., Arzumanyan, G. A., Obadi, M. K. A., Javanpour, A. A. & Liu, C. C. Scalable, Continuous Evolution of Genes at Mutation Rates above Genomic Error Thresholds. *Cell* **175**, 1946-1957.e13; 10.1016/j.cell.2018.10.021 (2018).

4. Esvelt, K. M., Carlson, J. C. & Liu, D. R. A system for the continuous directed evolution of biomolecules. *Nature* **472**, 499–503; 10.1038/nature09929 (2011).
5. Crook, N. *et al.* In vivo continuous evolution of genes and pathways in yeast. *Nature communications* **7**, 13051; 10.1038/ncomms13051 (2016).
6. Cravens, A., Jamil, O. K., Kong, D., Sockolovsky, J. T. & Smolke, C. D. Polymerase-guided base editing enables in vivo mutagenesis and rapid protein engineering. *Nature communications* **12**, 1579; 10.1038/s41467-021-21876-z (2021).
7. Halperin, S. O. *et al.* CRISPR-guided DNA polymerases enable diversification of all nucleotides in a tunable window. *Nature* **560**, 248–252; 10.1038/s41586-018-0384-8 (2018).
8. Zuo, E. *et al.* Cytosine base editor generates substantial off-target single-nucleotide variants in mouse embryos. *Science (New York, N.Y.)* **364**, 289–292; 10.1126/science.aav9973 (2019).
9. Jin, S. *et al.* Cytosine, but not adenine, base editors induce genome-wide off-target mutations in rice. *Science (New York, N.Y.)* **364**, 292–295; 10.1126/science.aaw7166 (2019).
10. McGrath, E. *et al.* Targeting specificity of APOBEC-based cytosine base editor in human iPSCs determined by whole genome sequencing. *Nature communications* **10**, 5353; 10.1038/s41467-019-13342-8 (2019).
11. Lee, H. K., Smith, H. E., Liu, C., Willi, M. & Hennighausen, L. Cytosine base editor 4 but not adenine base editor generates off-target mutations in mouse embryos. *Communications Biology* **3**, 19; 10.1038/s42003-019-0745-3 (2020).
12. Álvarez, B., Mencía, M., Lorenzo, V. de & Fernández, L. Á. In vivo diversification of target genomic sites using processive base deaminase fusions blocked by dCas9. *Nature communications* **11**, 6436; 10.1038/s41467-020-20230-z (2020).
13. Zacharioudakis, I. & Tzamaras, D. A novel CRE recombinase assay for quantification of GAL10-non coding RNA suppression on transcriptional leakage. *Biochemical and Biophysical Research Communications* **473**, 1191–1196; 10.1016/j.bbrc.2016.04.038 (2016).
14. Kar, R. K. *et al.* Stochastic galactokinase expression underlies GAL gene induction in a GAL3 mutant of *Saccharomyces cerevisiae*. *FEBS J* **281**, 1798–1817; 10.1111/febs.12741 (2014).
15. Thuronyi, B. W. *et al.* Continuous evolution of base editors with expanded target compatibility and improved activity. *Nature biotechnology* **37**, 1070–1079; 10.1038/s41587-019-0193-0 (2019).
16. Luria, S. E. & Delbrück, M. Mutations of Bacteria from Virus Sensitivity to Virus Resistance. *Genetics* **28**, 491–511; 10.1093/genetics/28.6.491 (1943).
17. Nile, A. H. *et al.* PITPs as targets for selectively interfering with phosphoinositide signaling in cells. *Nature Chemical Biology* **10**, 76–84; 10.1038/nchembio.1389 (2014).

Reviewers' Comments:

Reviewer #1:

Remarks to the Author:

The author's further research and answers convinced me.

Therefore, I recommend this paper to the journal Nature Communications.

Reviewer #2:

Remarks to the Author:

Zimmermen et al. have done a great job in addressing my review comments and improving their manuscript. I have no hesitation in recommending this for publication now. I expect many people will be excited to use their system, especially as it is optimised for the industrially-important yeast *S.cerevisiae*

REVIEWERS' COMMENTS

Reviewer #1 (Remarks to the Author):

The author's further research and answers convinced me.
Therefore, I recommend this paper to the journal Nature Communications.

We would like to thank the reviewer for his/her kind words.

Reviewer #2 (Remarks to the Author):

Zimmermen et al. have done a great job in addressing my review comments and improving their manuscript. I have no hesitation in recommending this for publication now. I expect many people will be excited to use their system, especially as it is optimised for the industrially-important yeast *S.cerevisiae*

We thank the reviewer for his/her thorough assessment of the paper and comments that helped to further improve the manuscript.

We are happy to see that both reviewers recommend our paper for publication.